



# Retrieving frozen ground surface temperature under the snowpack in Arctic permafrost area from SMOS observations

Juliette Ortet[1, 2, 3], Arnaud Mialon[1], Alain Royer[3, 4], Mike Schwank[5, 6], Manu Holmberg[7], Kimmo Rautiainen[7], Simone Bircher-Adrot[8], Andreas Colliander[9], Yann Kerr[1], and Alexandre Roy[2, 3]

[1]Univ Toulouse 3 Paul Sabatier, Univ Toulouse, CNES/IRD/CNRS/INRAe, CESBIO, Toulouse, France
[2]Département des sciences de l'environnement, Université du Québec à Trois-Rivières, Trois-Rivières, Quebec, G9A 5H7, Canada
[3]Centre d'études nordiques, Québec, Quebec, G1V 0A6, Canada
[4]Département de géomatique appliquée, Université de Sherbrooke, Sherbrooke, J1K 2R1, Canada
[5]Swiss Federal Institute for Forest, Snow and Landscape Research WSL, Switzerland
[6]Gamma Remote Sensing Research and Consulting Ltd., Switzerland
[7]Finnish Meteorological Institute, Earth Observation Research Unit, Finland
[8]MétéoSuisse, Payerne, Switzerland
[9]NASA Jet Propulsion Laboratory, California Institute of Technology, Pasadena, CA, USA

**Correspondence:** Juliette Ortet (juliette.ortet@uqtr.ca)

**Abstract.** We developed and evaluated a new method to retrieve ground surface temperatures $T_g$ below the snowpack from Soil Moisture and Ocean Salinity (SMOS) L-band brightness temperatures (BT). The study was performed over 21 reference sites providing with *in situ* ground temperatures $T_{g\text{-insitu}}$ in Northern Alaska from 2011 to 2020, representative of Arctic tundra underlined by continuous permafrost, and with various open water fractions. $T_g$ were obtained by inverting two types of microwave emission model (MEM) tailored for winter Arctic tundra environments. The first MEM assumed a homogeneous SMOS pixel and optimized the surface roughness $H_{r,gs}$. We observed the important influence of the frozen water bodies on $T_g$ retrievals. Accordingly, we used an advanced MEM that accounts for the water surfaces within the SMOS pixels and describes their emission using an optimized water-ice interface roughness parameter, $H_{r,wi}$. For sites with water fraction < 0.04, our methods (median correlation R = 0.60) outperformed the European Centre for Medium-Range Weather Forecasts reanalysis (ERA5) product (median R = 0.51) with respect to the reference sites. The bias between retrieved and *in situ* temperature was slightly negative (median bias = -0.2°C). For sites with water fraction > 0.20, our water fraction correction reduced the bias, but the correlation of the $T_g$ retrievals remained lower than that of ERA5. This study opens a new avenue for monitoring $T_g$ below the snowpack in the Arctic using L-band BT, by inversion of a relatively simple MEM and limited auxiliary data. Extending this study to the whole Arctic area and taking advantage of the 15 years of SMOS data to study spatio-temporal variability of winter $T_g$ in Arctic environments is excessively promising.

## 1 Introduction

The ground surface temperature $T_g$ is a key parameter for physical land surface processes. The observed increase in the surface air temperatures over the last decades (Druckenmiller and Jeffries, 2019) and $T_g$ (Biskaborn et al., 2019) in the Arctic regions



induced changes in land surface energy and water balance, impacting weather and climate at local and global scales (Schuur
et al., 2015; Chadburn et al., 2017; Turetsky et al., 2020). $T_{\mathrm{g}}$ changes also impact surface runoff and hydrological processes
(Rouse et al., 1997; Ala-Aho et al., 2021) and the ecosystem dynamics (Wang et al., 2019). In snow-covered conditions, $T_{\mathrm{g}}$
temporal dynamics are generally decoupled from air temperature (Bartlett et al., 2004; Cao et al., 2020) because of snow
thermal insulation capacity (Zhang, 2005; Domine et al., 2019). Hence, $T_{\mathrm{g}}$ modulates the permafrost active layer dynamics
and its spatial distribution (Dobiński, 2020). The Arctic freeze/thaw ground state associated with $T_{\mathrm{g}}$ is a key element of Arctic
climate change feedbacks as $T_{\mathrm{g}}$ is the main driver of $CO_2$ release through soil respiration during winter (Natali et al., 2019;
Mavrovic et al., 2023). However, meteorological stations over the Arctic are sparse and very few $T_{\mathrm{g}}$ observations are available
(Shiklomanov, 2012). Model and reanalysis data provide $T_{\mathrm{g}}$ at a global scale for decades but in Arctic areas, the results remain
uncertain (Royer et al., 2021b), mostly during winter when the Arctic is covered by snow (Herrington et al., 2024). Statistical,
empirical, and machine learning models (Aalto et al., 2018; Lembrechts et al., 2022; Guo et al., 2024) were proposed but the
insulation properties of snow coverage remain a major challenge to estimate $T_{\mathrm{g}}$ (Lembrechts et al., 2022).

Satellite remote sensing provides opportunities to map $T_{\mathrm{g}}$ in cold environments (Westermann et al., 2015). The land surface
temperature (LST) can be retrieved based on thermal radiometry (e.g. Jiménez-Muñoz et al. (2014)). However, during winter,
LST corresponds to the temperature of the snow surface (Westermann et al., 2012). High-frequency ($f > 10$ GHz microwave
data (Fily, 2003; Jones et al., 2007; André et al., 2015) showed limited results for determining the $T_{\mathrm{g}}$ under the snowpack
(Duan et al., 2020). Moreover, Köhn and Royer (2012) and Mialon et al. (2007) showed that when using AMSR-E and SSMI
observations, the derived LST corresponds to a thin layer (skin) at the air-snow interface. Marchand et al. (2018) showed
the potential of using passive microwaves to retrieve $T_{\mathrm{g}}$ by combining AMSR-E and MODIS data to inform a land surface
scheme. However, the study was performed in a unique site and the integration of remote sensing data in a land surface scheme
remains complex and operationally difficult to implement. It is well known that low microwave frequencies ($f < 10$ GHz) are
less sensitive to snow properties, and L-band (typical frequency $f = 1400$–$1427$ GHz, wavelength $\lambda \simeq 21$ cm) could provide
unique information about the frozen ground under the snow (Schwank et al., 2015; Lemmetyinen et al., 2016; Roy et al., 2017).

In this study, we developed a new approach to retrieve $T_{\mathrm{g}}$ under the snowpack in tundra environments from SMOS obser-
vations. The emitted radiations observed by SMOS are expressed in terms of brightness temperature (BT) and predominantly
determined by the effective temperature and the emissivity of the observed scene. By considering that the Arctic ground surface
remains frozen throughout winter, the ground emissivity remains constant and the BT depends mostly on $T_{\mathrm{g}}$. However, even
if ground emissivity remains constant, other contributions to the signal, including contributions from snow and water bodies,
should be considered in retrieving $T_{\mathrm{g}}$. We developed a microwave microwave emission model (MEM) for Arctic tundra condi-
tions to address the complex and heterogeneous scene observed at the SMOS footprint scale. The parameterization of central
components such as the frozen ground permittivity, the snow layer, and the fraction of snow and ice covered water bodies and
their impact on $T_{\mathrm{g}}$ retrievals were evaluated. The retrieved $T_{\mathrm{g}}$ were validated against *in situ* measurements from 21 sites across
northern Alaska and compared with the European Centre for Medium-Range Weather Forecasts (ECMWF) reanalysis (ERA5)
ground temperatures $T_{\mathrm{g\text{-}ERA5}}$ (Hersbach, H. et al., 2023).



## 2 Datasets

### 2.1 Brightness temperatures from SMOS

Operated by the European Space Agency (ESA), the SMOS satellite has been acquiring multi-angular BT at L-band since January 2010 (Kerr et al., 2010). We used the SMOS Level 3 brightness temperatures (L3BT) version 330 provided by the Centre Aval de Traitement des Données SMOS (CATDS) (CATDS, 2024). The L3BT are sampled on the global Equal Area Scalable Earth version 2.0 (EASE 2.0 grid, Brodzik et al. (2012)) using a cylindrical projection for daily ascending and descending orbits. Both vertical (V) and horizontal (H) polarizations are available for observation (off-nadir) angles $\theta$ from 0° to 60° binned

over 5-degree intervals (Al Bitar et al., 2017). The SMOS measurements are impacted by Radio Frequency Interferences (RFI) (Daganzo-Eusebio et al., 2013), whose consequences vary in time, so morning and afternoon orbits were considered separately. The revisit time is shorter than the three-day revisit at the equator and enables observations of the study area at least once a day. The BT are associated with the estimated radiometric accuracy and sample standard deviation obtained in the averaging of measurements into observation angle bins.

### 2.2 *In situ* measurements of ground temperatures


The 21 reference *in situ* sites are located across Alaska (US), in the Arctic region (Figure 1 and Table 1). The topography is flat and the continuous permafrost landscape integrates numerous lakes. Some sites are located close to the coast (Barrow, Lake 145, Fish Creek, Camden Bay) while others are disseminated inland. All the selected sites are located above the tree line and are representative of the tundra environment with vegetation characterized by low shrubs and mosses (Table 1). The study sites

are part of four different networks. The United States Geological Survey (USGS) (Urban, 2017) provided 14 sites from 1998 to 2019 as part of the Global Terrestrial Network for Permafrost (GTN-P). Three other sites come from the Carbon in Arctic Reservoirs Vulnerability Experiment (CARVE) (Oechel et al., 2016) between 2011 and 2015. The last four sites are part of the Soil Climate Analysis Network (SCAN) (Schaefer et al., 2007) and Snowpack Telemetry (SNOTEL) (Leavesley et al., 2010) and were accessed thanks to the International Soil Moisture Network (ISMN) (Dorigo et al., 2021). The *in situ* data is available

with an hourly temporal resolution and was selected from January 2011 to coincide with SMOS observations. For each site, ground temperatures ($T_{\text{g-insitu}}$) at variable probing depth are available (Table 1). Other variables such as air temperature at 2 m height and snow depth are available.





**Figure 1.** Distribution of the 21 ground-based $T_{\text{g-insitu}}$ stations used as a reference (background: the permafrost extent and tree line from Heginbottom et al. (2002). Sites coordinates are specified in Table 1.

## 2.3 Model reanalysis ground temperatures

The $T_{\text{g}}$ retrieved from the L3BT was compared to the fifth generation ECMWF re-analysis (ERA5) ground temperature product (Hersbach, H. et al., 2023). We used the shallower soil temperature (Level 1, 0 - 7 cm depth) $T_{\text{g-ERA5}}$ provided on a 0.25° resolution grid with an hourly temporal resolution.

## 2.4 Land cover

The land cover fraction was calculated from the ESA CCI L4 map at a 300 m spatial resolution, Version 2.0.7 (2015) (Defourny, P. et al., 2023). To obtain the fraction of a given land cover class for one grid cell, the number of ESA CCI pixels of the corresponding class was divided by the total number of ESA CCI pixels in a round buffer around the grid cell center. A 40 km

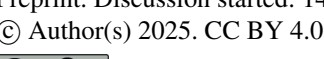



**Table 1.** *In situ* stations coordinates with the associated available probe depths. The land cover fractions extracted from the ESA CCI L4 map at 300 m, Version 2.0.7 (2015) (ESA) using a 40 km diameter buffer around the closest SMOS L3 grid cell center for each study site. Only classes with fractions above 5% for at least one site are are presented.

| Network | Site | Latitude in ° | Longitude in ° | Probe depth(s) in cm | Sh.[1] | Gr.[2] | Li.Mo.[3] | S.v.(15)[4] | Fl.[5] | B.a.[6] | W.[7] |
|---|---|---|---|---|---|---|---|---|---|---|---|
| CARVE Oechel et al. (2016) | Atqasuk | 70.47 | -157.409 | 5 | 0.00 | 0.00 | 0.38 | 0.04 | 0.26 | 0.03 | 0.24 |
| | Barrow | 71.323 | -156.597 | 5 | 0.00 | 0.01 | 0.40 | 0.13 | 0.14 | 0.00 | 0.32 |
| | Ivotuk | 68.486 | -155.748 | 5 | 0.01 | 0.31 | 0.19 | 0.42 | 0.00 | 0.05 | 0.00 |
| USGS Urban (2017) | Imigok | 69.98962 | -153.09384 | 5 to 120† | 0.00 | 0.00 | 0.57 | 0.16 | 0.00 | 0.00 | 0.23 |
| | Fish Creek | 70.33523 | -152.052 | 5 to 120† | 0.00 | 0.00 | 0.30 | 0.04 | 0.06 | 0.00 | 0.59 |
| | Umiat | 69.39568 | -152.14273 | 5 to 120† | 0.10 | 0.33 | 0.15 | 0.37 | 0.00 | 0.02 | 0.02 |
| | Tunalik | 70.19593 | -161.07812 | 5 to 120† | 0.02 | 0.27 | 0.33 | 0.32 | 0.02 | 0.01 | 0.02 |
| | Koluktak | 69.7516 | -154.61744 | 5 to 120† | 0.00 | 0.00 | 0.59 | 0.09 | 0.04 | 0.03 | 0.20 |
| | Niguanak | 69.88944 | -142.9845 | 5 to 120† | 0.00 | 0.01 | 0.22 | 0.50 | 0.04 | 0.01 | 0.22 |
| | Marsh Creek | 69.77762 | -144.79325 | 5 to 120† | 0.02 | 0.03 | 0.10 | 0.35 | 0.00 | 0.00 | 0.44 |
| | South Meade | 70.62847 | -156.83532 | 5 to 120† | 0.00 | 0.00 | 0.39 | 0.04 | 0.24 | 0.02 | 0.27 |
| | Camden Bay | 69.97196 | -144.77057 | 15 | 0.02 | 0.03 | 0.10 | 0.35 | 0.00 | 0.00 | 0.44 |
| | Awuna2 | 69.156 | -158.03005 | 15 | 0.03 | 0.74 | 0.02 | 0.19 | 0.00 | 0.00 | 0.00 |
| | Piksiksak | 70.03662 | -157.08137 | 5 to 120† | 0.17 | 0.16 | 0.42 | 0.15 | 0.02 | 0.00 | 0.04 |
| | East Teshekpuk | 70.56852 | -152.96498 | 5 to 120† | 0.00 | 0.00 | 0.38 | 0.05 | 0.15 | 0.01 | 0.41 |
| | Ikpikpuk | 70.44165 | -154.36563 | 5 to 120† | 0.00 | 0.00 | 0.45 | 0.11 | 0.08 | 0.03 | 0.32 |
| | Lake 145 | 70.6898 | -152.63325 | 15 | 0.00 | 0.00 | 0.42 | 0.05 | 0.13 | 0.01 | 0.39 |
| ISMN SNOTEL Leavesley et al. (2010) | Imnaviat Creek | 68.62 | -149.3 | 5 and 20 | 0.01 | 0.24 | 0.02 | 0.71 | 0.00 | 0.01 | 0.01 |
| | Kelly Station | 67.93 | -162.28 | 5 and 20 | 0.42 | 0.15 | 0.09 | 0.09 | 0.00 | 0.06 | 0.03 |
| | Atigun Pass | 68.13 | -149.48 | 5 and 20 | 0.02 | 0.03 | 0.34 | 0.33 | 0.00 | 0.24 | 0.01 |
| ISMN SCAN Schaefer et al. (2007) | Ikalukrok Creek | 68.08 | -163.0 | 5 and 20 | 0.10 | 0.25 | 0.09 | 0.42 | 0.00 | 0.11 | 0.00 |

† "5 to 120" refers to all the available depths for the USGS sites, i.e. 5 – 10 – 15 – 20 – 25 – 30 – 45 – 70 – 95 – 120 cm.
[1] Shrubland  [2] Grassland  [3] Lichens and mosses  [4] Sparse vegetation (tree shrub herbaceous cover) (<15%)  [6] Bare areas  [7] Water bodies
[5] Shrub or herbaceous cover flooded fresh/saline/barkish water



diameter buffer zone around each SMOS L3 grid cell center roughly corresponds to a 3 dB antenna pattern cut-off assimilated to the instrumental spatial resolution. The water fraction at each site was within a 40 km buffer. The land cover classes were used for the *in situ* environment characterization and the analysis of the results. The land cover fractions are summed up in Table 1. None of the sites are significantly covered by trees or high vegetation.

## 3 Methods

### 3.1 Pre-processing

Our retrievals were based on L-band $T_{\mathrm{B}}$ in H and V polarizations and at angles from 0 to 60°. The $T_{\mathrm{B}}$ were filtered if the RFI ratio (defined as the sum of the RFI flagged instances divided by the sum of the SMOS L1 views combined in each of the L3BT 5-degree angle bin) was more than 0.1. Due to the RFI situation in North America (Aksoy and Johnson, 2013), observations before 2012 were discarded. In winter, $T_{\mathrm{g}}$ under the snowpack is expected to be diurnally relatively stable (Bartlett et al., 2004). Consequently, we only focused on the daily morning (ascending) orbit passes (approx. 6 a.m local overpass). We used the $T_{\mathrm{g\text{-}insitu}}$ at 5 cm depth to focus on the same ground surface layer for all sites. An exception was made for Awuna2, Camden Bay, and Lake 145 where only 15 cm depth measurements were available. For each L3BT, we selected the closest $T_{\mathrm{g\text{-}insitu}}$ observed within 30 minutes of the mean satellite overpass time. The retrieval was performed only when $T_{\mathrm{g\text{-}insitu}} < -5°\mathrm{C}$ to ensure that ground conditions satisfy our stable frozen ground permittivity hypothesis (Pardo Lara et al., 2020). We also compared $T_{\mathrm{g\text{-}ERA5}}$ with respect to $T_{\mathrm{g\text{-}insitu}}$. For each site, we considered the nearest neighbor ERA5 node and used the closest time to the satellite overpass time.

### 3.2 Microwave emission model for the Arctic tundra during winter

Our proposed approach for $T_{\mathrm{g}}$ retrieval required an inversion model based on a MEM (Figure 2). The upwelling surface $T_{\mathrm{B,surf}}^{p}(\theta)$ was considered to be the linear combination of the upwelling BT from the snow-covered ground $T_{\mathrm{B,G}}^{p}(\theta)$, from the snow and ice covered water bodies $T_{\mathrm{B,WI}}^{p}(\theta)$ weighted by the water bodies fraction $\nu_{\mathrm{wi}}$:

$$T_{\mathrm{B,surf}}^{p}(\theta) = (1 - \nu_{\mathrm{wi}}) \cdot T_{\mathrm{B,G}}^{p}(\theta) + \nu_{\mathrm{wi}} \cdot T_{\mathrm{B,WI}}^{p}(\theta) \tag{1}$$

$T_{\mathrm{B,G}}^{p}(\theta)$ and $T_{\mathrm{B,WI}}^{p}(\theta)$ were simulated with multi-layer configurations of the Two-Stream model (Schwank et al., 2014) and the Microwave Emission Model of Layered Snowpacks (MEMLS) (Mätzler and Wiesmann, 2012) reflecting the two emission model scenarios depicted in Figure 2. $T_{\mathrm{B,G}}^{p}(\theta)$ resulted from a submodel considering the snow and the atmosphere as two horizontal layers atop the ground which is an infinite half-space. Note that the low vegetation of the tundra is not considered in the submodel. In the case of $T_{\mathrm{B,WI}}^{p}(\theta)$, the submodel is made of three horizontal layers (ice, snow, and atmosphere) above the water as an infinite half-space. The layers and infinite half-spaces parametrizations are described in the following Sections (Sections 3.2.1, 3.2.2, 3.2.3). $T_{\mathrm{B,G}}^{p}(\theta)$ and $T_{\mathrm{B,WI}}^{p}(\theta)$ were also corrected from the atmosphere opacity $\tau_{\mathrm{atm}}(\theta)$. The deep sky and atmosphere upwelling and downwelling contributions were taken into account as in (Kerr et al., 2020), depending on





$T_{\mathrm{B,sky}}$, $T_{\mathrm{B,atm}}(\theta)$ and $\tau_{\mathrm{atm}}(\theta)$ (Table 2).

Our MEM considered microwave interactions at the interface between two layers: the reflectivity and the refractivity. The reflectivities of the smooth surface between layer $n$ and $n+1$ are noted as $s^{\mathrm{H}*}(\theta)$ and $s^{\mathrm{V}*}(\theta)$ and were given by the Fresnel
reflection coefficients (Ulaby and Long, 2014):

$$s^{\mathrm{H}*}(\theta) = \left| \frac{\sqrt{\varepsilon_n} \cdot A - \sqrt{\varepsilon_{n+1}} \cdot B}{\sqrt{\varepsilon_n} \cdot A + \sqrt{\varepsilon_{n+1}} \cdot B} \right|^2 \qquad\qquad s^{\mathrm{V}*}(\theta) = \left| \frac{\sqrt{\varepsilon_{n+1}} \cdot A - \sqrt{\varepsilon_n} \cdot B}{\sqrt{\varepsilon_{n+1}} \cdot A + \sqrt{\varepsilon_n} \cdot B} \right|^2 \qquad (2)$$

with $\quad A = \cos(\theta_n) \quad$ and $\quad B = \sqrt{1 - (1-A^2) \cdot \dfrac{\varepsilon_n}{\varepsilon_{n+1}}}$

where H and V stand for horizontal and vertical polarization, $\theta$ account for the incidence angle and $\varepsilon_n$ is the layer $n$ complex
dielectric constant.

The H-Q-N model (Wang and Choudhury, 1981) was proposed to empirically consider surface effects (including roughness) in the reflectivity and can be expressed as:

$$s^p(\theta) = \left[(1-Q_{\mathrm{r}})s_n^{p*}(\theta) + Q_{\mathrm{r}}s^{q*}(\theta)\right] \cdot \exp\left(-H_{\mathrm{r}}\cos^{N_{\mathrm{r}}^p}(\theta)\right) \qquad (3)$$

where $p$ and $q$ are the two polarizations ($q$ is H (resp. V) when $p$ is V (resp. H)). The surface effects were taken into account
with four parameters: the polarization mixing ratio $Q_{\mathrm{r}}$, the angular effect parameters $N_{\mathrm{r}}^{\mathrm{H}}$, and $N_{\mathrm{r}}^{\mathrm{V}}$ and the effective roughness parameter $H_{\mathrm{r}}$. These four parameters account for not only the geometric roughness effects but also the spatial heterogeneity of the surface characteristics. For instance, Escorihuela et al. (2007) showed a $H_{\mathrm{r}}$ dependence on soil moisture content for a ground-air interface. Our values for those parameters are detailed in the following sections and summed up in Table 2.

The angle deviation due to refractivity at the interface between the layers $n$ and $n+1$ is given by Snell-Descartes law:

$$\theta_n = \arcsin\left(\sqrt{\frac{\varepsilon_{n+1}}{\varepsilon_n}} \sin\theta_{n+1}\right) \qquad (4)$$

where $\varepsilon_n$ is the layer $n$ complex dielectric constant Ulaby et al. (1984).

### 3.2.1 Frozen ground parametrization

The bottom-most infinite half-space representing the ground was described using the following parameters: $T_{\mathrm{g}}$, $\varepsilon_{\mathrm{frozen}}$, $H_{\mathrm{r,gs}}$,
$Q_{\mathrm{r,gs}}$, $N_{\mathrm{r,gs}}^p$ (see Figure 2). The ground-snow interface reflectivity $s_{\mathrm{gs}}^p$ was obtained from equations 2 and 3. This study aimed to retrieve the ground surface temperature $T_{\mathrm{g}}$ by considering a fixed and constant ground permittivity in frozen conditions. Various models describe the ground permittivity at 1.4 GHz (Mironov et al., 2009; Bircher et al., 2016; Park et al., 2017), but very few in the case of frozen ground (Hallikainen et al., 1985; Mironov et al., 2015). The permittivity of a frozen ground was set to $\varepsilon_{\mathrm{frozen}} = 5.0 + 0.5\,\mathrm{i}$, similar to past studies (Schwank et al., 2014; Holmberg et al., 2024) and SMOS algorithm (Kerr
et al., 2020). We considered the ground surface reflectivity as in Equation 3 accounting for various effects including roughness



using four parameters ($H_{\mathrm{r,gs}}$, $Q_{\mathrm{r,gs}}$, $N_{\mathrm{r,gs}}^{\mathrm{H}}$ and $N_{\mathrm{r,gs}}^{\mathrm{V}}$). The polarization mixing ratio $Q_{\mathrm{r,gs}}$ (Wang and Choudhury, 1981) as well as the angular effects parameters $N_{\mathrm{r,gs}}^{\mathrm{H}}$ and $N_{\mathrm{r,gs}}^{\mathrm{V}}$) were set to 0, as suggested by several studies (Kerr et al., 2020; Wigneron et al., 2011; Lawrence et al., 2013). $H_{\mathrm{r,gs}}$ value was optimized for all the sites using a range of 0 to 1 with 0.1 increments.

### 3.2.2 Dry snow parametrization

The layer accounting for the snow was defined by its effective temperature $T_{\mathrm{s}}$, its permittivity $\varepsilon_{\mathrm{s}}$, and the layer internal transmissivity $t_{\mathrm{s}}$ and reflectivity $r_{\mathrm{s}}$ (Figure 2). According to Schwank et al. (2015) and Rautiainen et al. (2016), dry snow can be considered transparent at L-band, i.e. its internal transmissivity and reflectivity are $t_{\mathrm{s}} = 1$ and $r_{\mathrm{s}} = 0$. Consequently, our model became independent of $T_{\mathrm{s}}$. However, Schwank et al. (2015) showed that air-snow interface impacts on impedance matching can not be ignored, i.e. the snow surface reflectivity $s_{\mathrm{s}}^{p*} \neq 0$. We considered refraction (Equation 4) and reflection for a smooth
air-snow interface (Equation 2). The dry snow permittivity was set to $\varepsilon_{\mathrm{s}} = 1.53$ according to Equation 4 of Schwank et al. (2015) for a mean snow density $\rho_{\mathrm{s}} = 300$ kg m$^3$, which corresponds to the high Arctic snowpack average density observed by Derksen et al. (2014) and Roy et al. (2017). We assume a snowpack with the same parameters above the ground and the ice-covered water bodies.

### 3.2.3 Snow and ice covered water bodies parametrization

During winter, water bodies are fully covered by an ice layer with liquid water remaining below the ice layer (Adams and Lasenby, 1985; Jeffries et al., 2013). The ice layer was defined by its permittivity $\varepsilon_{\mathrm{i}} = 3.18$ (Mätzler, 2006) and considered transparent (internal transmissivity $t_i = 1$ and internal reflectivity $r_i = 0$). However, smooth surface refraction (Equation 4) and reflection $s_{\mathrm{is}}^{p*}$ (Equation 2) were taken into account at the ice-snow interface. Similarly to the ground layer, the liquid water layer was defined with $T_{\mathrm{w}}$, $\varepsilon_{\mathrm{w}}$, $H_{\mathrm{r,wi}}$, $Q_{\mathrm{r,wi}}$ and $N_{\mathrm{r,wi}}^{p}$ (Figure 2). The water temperature $T_{\mathrm{w}}$ was considered constant
throughout winter and equal to 2°C (Oveisy et al., 2012). We consider fresh water whose L-band permittivity $\varepsilon_{\mathrm{w}}$ was fixed to $86 + 13\,\mathrm{i}$ (Liebe et al., 1991; Mätzler, 2006; Ulaby and Long, 2014). The water-ice interface reflectivity $s_{\mathrm{wi}}^{p}$ was obtained from equation 3, accounting for the water-ice interface heterogeneity. $Q_{\mathrm{r,wi}}$, $N_{\mathrm{r,wi}}^{\mathrm{H}}$ and $N_{\mathrm{r,wi}}^{\mathrm{V}}$ were set to 0 (Choudhury et al., 1979). $H_{\mathrm{r,wi}}$ value was optimized for all the sites on a range of 0 to 2 with an iteration step of 0.1. The water body $\nu_{\mathrm{wi}}$ accounted for the area percentage of the considered SMOS node covered by water bodies based on the water class from ESA CCI landcover
(Table 1).

### 3.2.4 Microwave emission model configurations

Figure 2 depicts a schematic of the MEMs and Table 2 summarizes the input parameters. This study tested two configurations: one considering a homogeneous scene with only ground (hereafter named MEM$_{\mathrm{G}}$) and one with a heterogeneous scene composed of ground and snow and ice covered water bodies (hereafter named MEM$_{\mathrm{G+WI}}$).



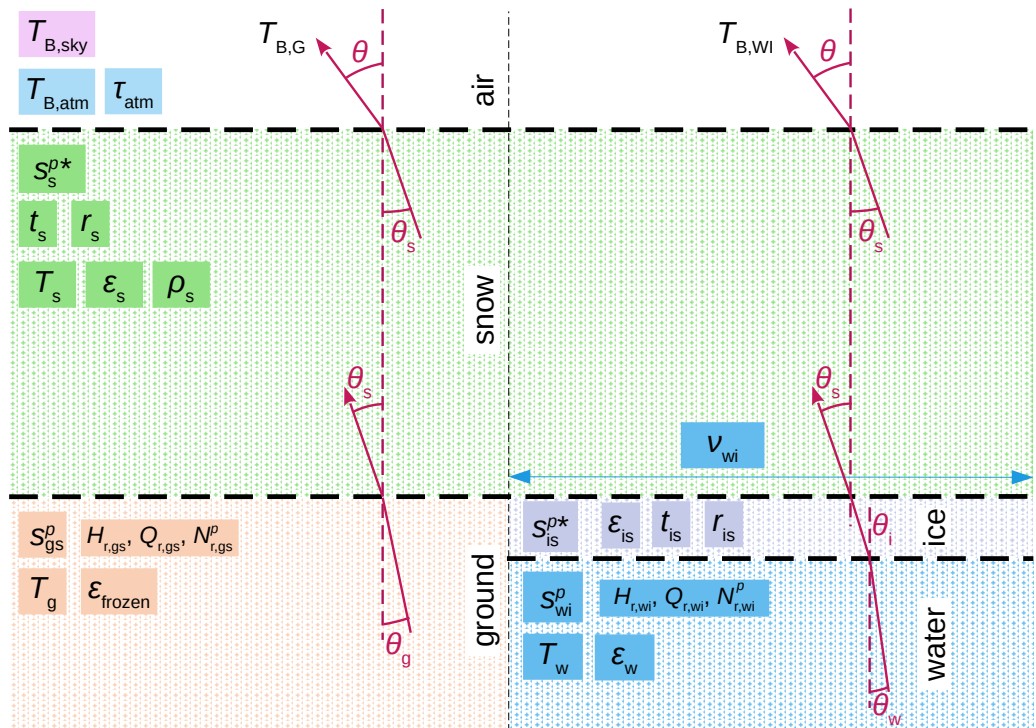

**Figure 2.** Schematic representation of the MEMs for modeling a winter tundra scene at L-band.
$\mathrm{MEM_G}$ only considers the left side of the sketch, $\mathrm{MEM_{G+WI}}$ considers both sides.



**Table 2.** Input parameters values of the MEM for modeling a winter tundra scene at L-band.

| Layer | Parameter | Description | Value |
|---|---|---|---|
| Atmosphere | $T_{\mathrm{B,sky}}$ | Deep sky BT | 2.7 K |
| | $T_{\mathrm{B,atm}}$ | Atmosphere BT | 2.2 K at nadir[†] |
| | $\tau_{\mathrm{atm}}$ | Atmosphere opacity | 0.01 at nadir[†] |
| Snow | $s_{\mathrm{s}}^{p*}$ | Snow-air interface reflectivity | Equation 2 |
| | $t_{\mathrm{s}}$ | Snow internal transmissivity | 1 |
| | $r_{\mathrm{s}}$ | Snow internal reflectivity | 0 |
| | $\varepsilon_{\mathrm{s}}$ | Dry snow permittivity | 1.53 |
| | $\rho_{\mathrm{s}}$ | Mean snow density | 300 kg m$^{-3}$ |
| Ground | $s_{\mathrm{gs}}^{p}$ | Ground-snow reflectivity | Equation 3 |
| | $H_{\mathrm{r,gs}}$ | Ground roughness | [0-1] |
| | $Q_{\mathrm{r,gs}}$ | Ground polarization ratio | 0 |
| | $N_{\mathrm{r,gs}}^{\mathrm{H}}$ | Ground angular dependent effects (in H) | 0 |
| | $N_{\mathrm{r,gs}}^{\mathrm{V}}$ | Ground angular dependent effects (in V) | 0 |
| | $\varepsilon_{\mathrm{frozen}}$ | Frozen ground permittivity | $5 + 0.5\,\mathrm{i}$ |
| | $T_{\mathrm{g}}$ | Effective ground temperature | Retrieved |
| Water body | $\nu_{\mathrm{wi}}$ | Water body fraction | 0 or Table 1 |
| | $s_{\mathrm{is}}^{p*}$ | Ice-snow reflectivity | Equation 2 |
| | $r_{\mathrm{i}}$ | Ice internal reflectivity | 0 |
| | $t_{\mathrm{i}}$ | Ice internal transmissivity | 1 |
| | $\varepsilon_{\mathrm{i}}$ | Ice permittivity | 3.18 |
| | $s_{\mathrm{wi}}^{p}$ | Water body-ice reflectivity | Equation 3 |
| | $H_{\mathrm{r,wi}}$ | Water body roughness | [0-1] |
| | $Q_{\mathrm{r,wi}}$ | Water body polarization ratio | 0 |
| | $N_{\mathrm{r,wi}}^{\mathrm{H}}$ | Water body angular dependent effects (in H) | 0 |
| | $N_{\mathrm{r,wi}}^{\mathrm{V}}$ | Water body angular dependent effects (in V) | 0 |
| | $\varepsilon_{\mathrm{w}}$ | Water permittivity | $86 + 13\,\mathrm{i}$ |
| | $T_{\mathrm{w}}$ | Water temperature | 2°C |

[†] Example value for $\theta = 0°$. For all the angles, $T_{\mathrm{B,atm}}$ and $\tau_{\mathrm{atm}}$ are calculated as in Kerr et al. (2020).





### 3.3 Cost function for frozen ground temperature retrievals

Both $\text{MEM}_\text{G}$ and $\text{MEM}_\text{G+WI}$ described in Section 3.2 were inverted to retrieve the frozen ground temperature $T_\text{g}$, by minimizing the following cost function:

$$\text{CF}(T_\text{g}) = \sum_{p,\theta_k} \left( \frac{T^p_{\text{B,obs}}(\theta_k) - T^p_{\text{B,sim}}(\theta_k, T_\text{g})}{\sigma T^p_\text{B}(\theta_k)} \right)^2 \tag{5}$$

where $T^p_{\text{B,obs}}(\theta_k)$ and $T^p_{\text{B,sim}}(\theta_k, T_\text{g})$ are the observed and simulated BT for both H and V polarizations and at various incidence angle bins $\theta_k$. The BT standard deviation $\sigma T^p_\text{B}(\theta_k)$ is computed from the estimated radiometric accuracy and sample standard deviation obtained in the averaging of measurements into observation angle bin $k$.

### 3.4 Post-processing

The first aim of the post-processing was to reduce the influence of outliers. The retrieved $T_\text{g}$ below the first 1% quantile and above the last 99% quantile of each site were considered outliers and discarded. We removed the $T_\text{g-ERA5}$ at these dates in the ERA5 time series to ensure that we compared a data pull with the same size. A low short-term variability is expected between $T_\text{g}$ under the snowpack that acts like a thermal insulator. The final step smoothed the $T_\text{g}$ time series to reduce the impact of the noise in SMOS BT to the retrievals. We used a z-score smoothing, to limit the variations of $T_\text{g}$ to 1 standard deviation for a 5-day window. At a date $t$, the local average $\overline{T^t_\text{g}}$ and standard deviation $\sigma\left(T^t_\text{g}\right)$ are calculated for a 5-day window around each $T^t_\text{g}$. If $T^t_\text{g} > \overline{T^t_\text{g}} + 1 \cdot \sigma\left(T^t_\text{g}\right)$, $T^t_\text{g}$ is replaced by $\overline{T^t_\text{g}}$.

### 3.5 Metrics

Three statistical indicators were used to assess the comparison between the retrieved $T_\text{g}$ and the reference temperatures $T_\text{g-insitu}$ ((Entekhabi et al., 2010; Gruber et al., 2020)). The unbiased Root Mean Square Deviation (ubRMSD) is used for uncertainty estimation as it is corrected from the bias between the two time series (Kerr et al., 2016a; Benninga et al., 2020). The bias corresponds to the mean difference between the compared time series of $T_\text{g}$ and $T_\text{g-insitu}$. The Pearson correlation coefficient (R) accounts for the similarities in temporal dynamics of the two time series. Each metric was computed for the whole time series for each site and was provided with its confidence intervals (CI) at 5 and 95%. Analytical solutions enabled us to find the CI of the bias, the ubRMSD and the R (Gruber et al., 2020). We also evaluated $T_\text{g-ERA5}$ with respect to $T_\text{g-insitu}$ with similar metrics.





# 4 Results

## 4.1 Parameters optimization evaluation

### 4.1.1 $H_{\mathrm{r,gs}}$ optimization

In the $\mathrm{MEM_G}$ configuration, we retrieved $T_{\mathrm{g}}$ by testing $H_{\mathrm{r,gs}}$ values from 0 to 1 with 0.1 increments. Figure 3 shows the biases obtained with all tested $H_{\mathrm{r,gs}}$ and biases obtained with $T_{\mathrm{g\text{-}ERA5}}$ for each site, with respect to $T_{\mathrm{g\text{-}insitu}}$. For all sites, the bias changed in the negative direction with increasing $H_{\mathrm{r,gs}}$. For sites with $\nu_{\mathrm{wi}} \leq 0.04$, the biases went from positive down to 205 negative values with increasing $H_{\mathrm{r,gs}}$, except for Awuna2 and Umiat whose biases remained positive. For sites with $\nu_{\mathrm{wi}} \geq 0.20$, the biases of numerous sites remained negative and went down close to -30°C. This suggests that the water bodies strongly impact the $T_{\mathrm{g}}$ retrieval bias. That is why we optimized the value of $H_{\mathrm{r,gs}}$ only on sites less affected by water bodies. For sites with $\nu_{\mathrm{wi}} \leq 0.04$, the bias was minimized with $H_{\mathrm{r,gs}} = 0.8$ (average = 0.2°C, median = -0.2°C, Q1 = -1.6°C, Q3 = 0.8°C, range = 2.4°C). Surprisingly, the sites with the highest $\nu_{\mathrm{wi}}$ (between 0.44 and 0.59) showed positive biases for some $H_{\mathrm{r,g}}$.

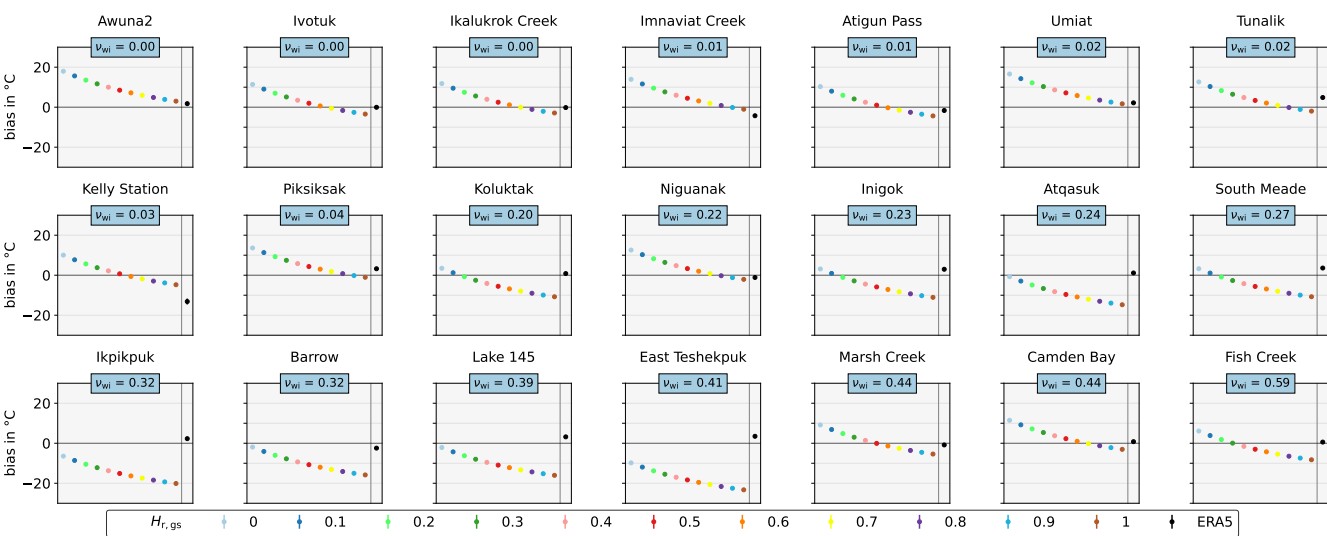

**Figure 3.** Bias per site for each $H_{\mathrm{r,gs}}$ used in the inversion with the $\mathrm{MEM_G}$ model. Each graph corresponds to one site. $H_{\mathrm{r,gs}}$ values are represented by a unique color and are ranged from 0 to 1 on the x-axis. The last point of each graph, in black, is obtained with $T_{\mathrm{g\text{-}ERA5}}$. The y-axis corresponds to the bias $T_{\mathrm{g}} - T_{\mathrm{g\text{-}insitu}}$. Each point is symbolized with error bars that correspond to the confidence interval. The sites are ordered in ascending order of water fraction ($\nu_{\mathrm{wi}}$ in the light blue box).

### 4.1.2 $H_{\mathrm{r,wi}}$ optimization

The results in Section 4.1.1 showed that the $T_{\mathrm{g}}$ retrieval bias strongly depends on water fraction. The $\mathrm{MEM_{G+WI}}$ model accounted for the presence of frozen water bodies (i.e. $\nu_{\mathrm{wi}} \neq 0$) in the $T_{\mathrm{B}}$ calculation (Figure 2). In this configuration, $T_{\mathrm{g}}$ was retrieved with different tested $H_{\mathrm{r,wi}}$ values from 0 to 1 with 0.1 increments. $H_{\mathrm{r,gs}}$ was set to 0.8 as shown in Section





4.1.1. For each site, Figure 4 shows the biases obtained with various $H_{\mathrm{r,wi}}$ and compared with $T_{\mathrm{g\text{-}ERA5}}$ bias with respect to
$Tg\text{-}insitu$. The higher $H_{\mathrm{r,wi}}$ the more negative the bias, while slope of the variations is linked to $\nu_{\mathrm{wi}}$. As expected, for sites
with $\nu_{\mathrm{wi}} \leq 0.04$, the biases showed little variations for all $H_{\mathrm{r,wi}}$. At Piksiksak ($\nu_{\mathrm{wi}} = 0.04$) bias went from 5.2°C ($H_{\mathrm{r,wi}} = 0$)
down to 2.0°C $H_{\mathrm{r,wi}} = 1$. For sites with $\nu_{\mathrm{wi}} \geq 0.20$, the biases highly varied with increasing $H_{\mathrm{r,wi}}$. For instance at Atqasuk
($\nu_{\mathrm{wi}} = 0.24$), the bias decreased from 16.5°C to -7.8°C with $H_{\mathrm{r,wi}} = 0$ and $H_{\mathrm{r,wi}} = 1$. At East Teshekpuk ($\nu_{\mathrm{wi}} = 0.41$), the
bias for the $H_{\mathrm{r,wi}}$ extrema decreased from 37.0°C to -16.7°C. For the sites with the highest $\nu_{\mathrm{wi}}$ (between 0.44 and 0.59), all the
biases remained larger than 15°C for the tested $H_{\mathrm{r,wi}}$ range. Consequently, we do not consider them in the following analysis
of the water body correction method. For the sites with $0.20 \leq \nu_{\mathrm{wi}} \leq 0.41$, the bias was minimized with $H_{\mathrm{r,wi}} = 0.7$ (average
= 0.7°C, median = 0.2°C, Q1 = -2.9°C, Q3 = 2.8°C, range = 5.7°C).

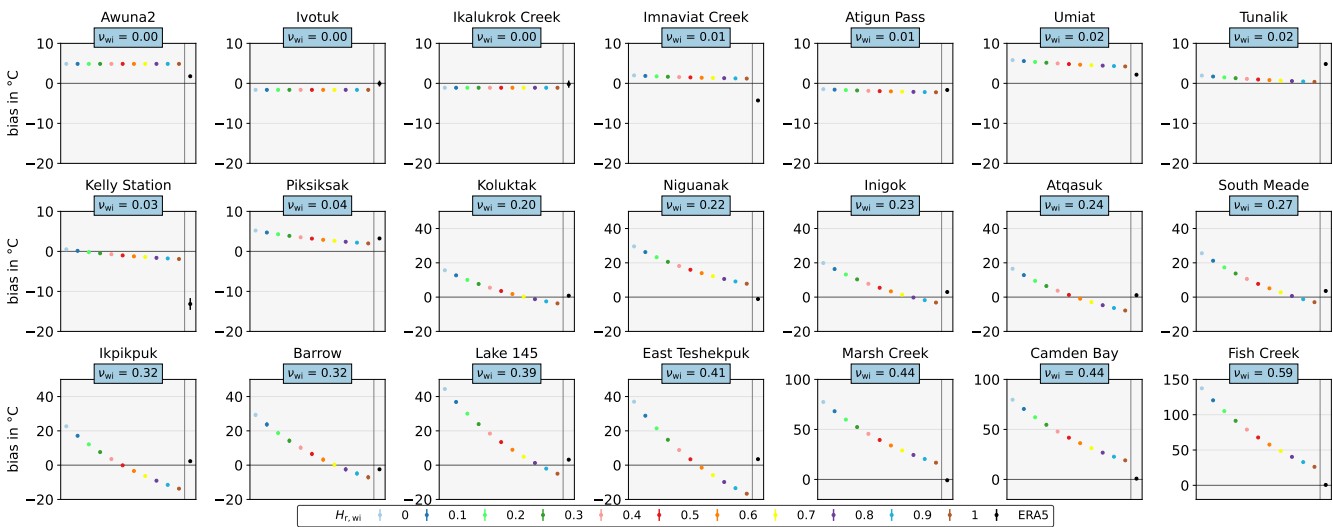

**Figure 4.** Bias per site for each $H_{\mathrm{r,wi}}$ used in the inversion. Each graph corresponds to one site. $H_{\mathrm{r,wi}}$ values are represented by a unique
color and marker combination (see Legend) and are ranged from 0 to 1 with a 0.1 step on the x-axis. The last point of each graph, in black,
is obtained with $T_{\mathrm{g\text{-}ERA5}}$. The y-axis corresponds to the bias $T_{\mathrm{g}} - T_{\mathrm{g\text{-}insitu}}$. Note that the y-axis scale is variable. Each point is symbolized
with error bars that correspond to the 5-95% confidence interval. The sites are ordered in ascending order of water fraction ($\nu_{\mathrm{wi}}$ in the light
blue box).

## 4.2   $T_{\mathbf{g}}$ retrievals evaluation

### 4.2.1   $T_{\mathbf{g}}$ retrievals for sites with $\nu_{\mathbf{wi}} \leq 0.04$

The R, bias, and ubRMSD using $\mathrm{MEM_G}$ with $H_{\mathrm{r,gs}} = 0.8$ and $\mathrm{MEM_{G+WI}}$ with $H_{\mathrm{r,wi}} = 1$ were compared to $T_{\mathrm{g\text{-}ERA5}}$ metrics
in Figure 5. For the sites with $\nu_{\mathrm{wi}} \leq 0.04$, when accounting for the water bodies with $\mathrm{MEM_{G+WI}}$, we selected $H_{\mathrm{r,wi}} = 1$ for
the ice-water interface as it minimized the bias average of these sites (average = 0.6°C). Each metric (in grey) is given with
its confidence limits at 5% (orange) and 95% (blue). This representation enables us to show the dispersion of the metrics for





all the considered sites. The R values of the retrieved $T_g$ (median = 0.60 for both MEM$_G$ and MEM$_{G+WI}$) were better than
ERA5 (median = 0.51). Moreover, in the case of ERA5, the interquartile range was larger (Q1 = 0.33, Q3 = 0.55, range = 0.22)
and the 5% confidence limit went down negative values. All the biases are centered around zero (mean = 0.2°C for MEM$_G$,
0.6°C for MEM$_{G+WI}$ and -0.8°C for ERA5), and all the absolute biases were lower than 5°C, except an outlier for ERA5 with
a strong negative bias = -13.1°C (Kelly Station, according to Figure 3). The ubRMSD from both inversions (median = 2.1°C
for both MEM$_G$ and MEM$_{G+WI}$) were significantly smaller than the ones from ERA5 (median = 3.9°C).

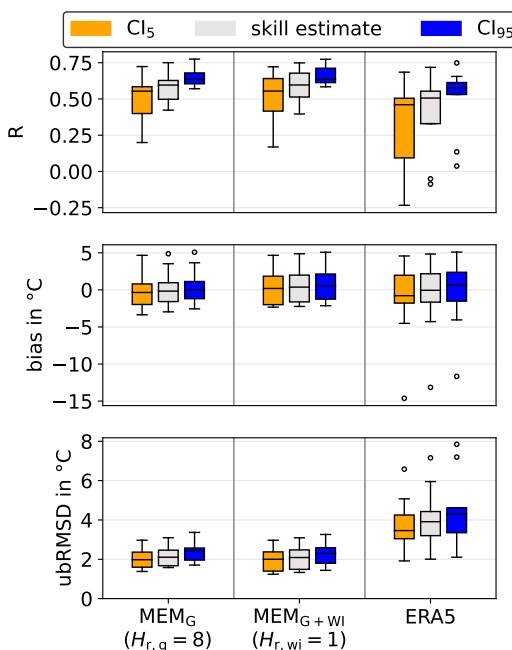

**Figure 5.** Summary statistics of R, bias and ubRMSD for sites with $\nu_{wi} \leq 0.04$. The boxes show the median and interquartile range and
whiskers show the 5 and 95 percentiles obtained from all the considered sites. The grey box corresponds to the skill estimate (R, bias, or
ubRMSD). Respectively, the orange and blue boxes correspond to the associated 5% and 95% confidence interval limits obtained from all
the considered sites. The x-axis corresponds to the $H_{r,wi}$ used in the inversion. The boxes are respectively obtained from: MEM$_G$ with $H_{r,gs}$
= 0 (left), MEM$_{G+WI}$ with $H_{r,wi}$ = 1 (center) and ERA5 (right).

### 4.2.2 $T_g$ retrievals for sites with $0.20 \leq \nu_{wi} \leq 0.41$

The overall R, bias and ubRMSD for MEM$_{G+WI}$ with different $H_{r,wi}$ are summarized in Figure 6 with the corresponding
MEM$_G$ (with $H_{r,gs}$ = 0.8) and ERA5 metrics. Similarly to Figure 5, the 5% (orange) and 95% (blue) confidence intervals are
given with each metric (in grey) and the boxes show the metrics dispersion. The R values remained the same for all $H_{r,wi}$ and
equal to the R reached with MEM$_G$ (median R = 0.21), but lower than ERA5 (median R = 0.62). The biases went more negative
with increasing $H_{r,wi}$ values. The bias was minimized for $H_{r,wi}$ = 0.7 (Section 4.1.2), with a median value (0.2°C) which was
closer to 0 than the bias with MEM$_G$ (median = -13.0°C) and with ERA5 (median = 2.3°C). Yet, for bias, the interquartile





range for $H_{r,wi} = 0.7$ (Q1 = -2.9°C, Q3 = 2.8°C, range = 5.7°C) remained much larger than ERA5 (Q1 = 0.8°C, Q3 = 3.2°C, range = 2.4°C), which meant that the bias remained higher for some of the sites. A wider range (Q1 = 4.4°C, Q3 = 6.6°C, range = 2.2°C) was also observed for the ubRMSD for all the $H_{r,wi}$ and $MEM_G$ (Q1 = 3.7°C, Q3 = 5.3°C, range = 1.6°C) with respect to ERA5 (Q1 = 3.2°C, Q3 = 3.5°C, range = 0.2°C).

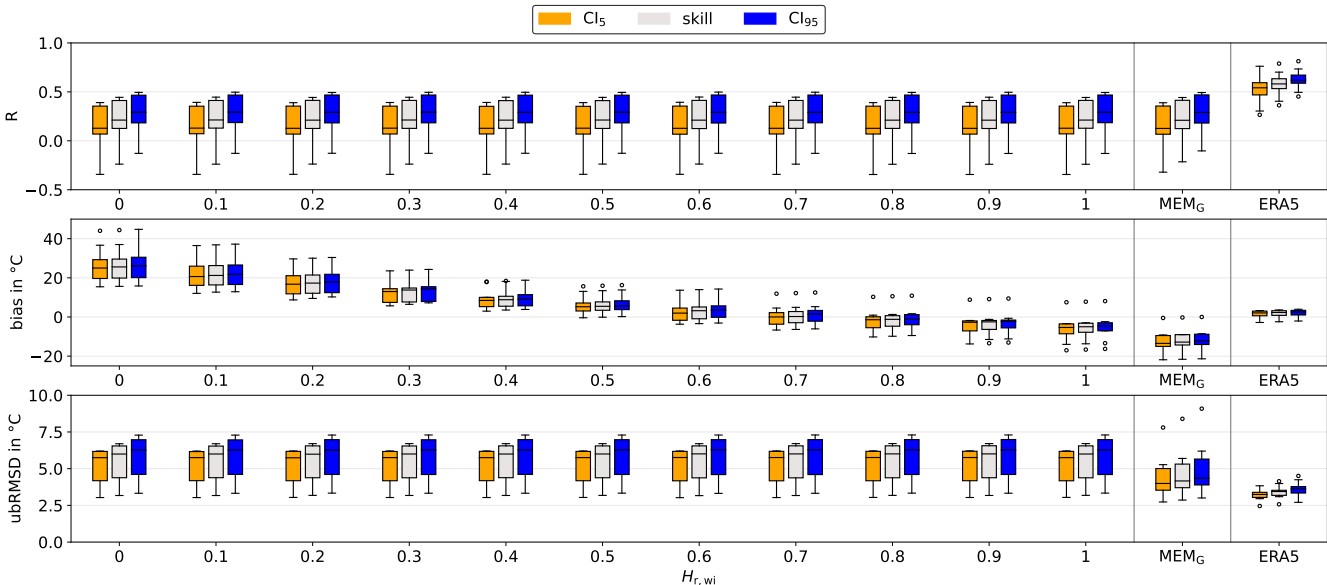

**Figure 6.** Summary statistics (in grey) of R, bias and ubRMSD and their 5% (in orange) and 95% (in blue) confidence intervals for sites with $0.20 \leq \nu_{wi} \leq 0.41$. Boxes represent the site median and interquartile range ($Q_3$ - $Q_1$) and whiskers represent the 5 and 95 percentiles. The x-axis corresponds to the $H_{r,wi}$ used in the inversion. The rightmost boxes are obtained with ERA5.

## 5 Discussion

The SMOS satellite was originally designed to focus on soil moisture and ocean salinity, but the applications extend to biomass monitoring (Kerr et al., 2010, 2016b; Mialon et al., 2020) and soil freeze-thaw state (Rautiainen et al., 2014, 2016). Recently, cryosphere applications have been increasingly investigated (Leduc-Leballeur et al., 2020; Schwank et al., 2021; Holmberg et al., 2024). The synergy between theses studies should be further explored. For instance, producing $T_g$ maps over the Arctic could complement the information from the freeze-thaw state products.

The retrieval model parametrization evaluation showed clear contrasting results according to the water bodies' fraction over sites. $T_g$ retrievals outperformed ERA-5 when $\nu_{wi} \leq 0.04$ but are mitigated when $\nu_{wi} \geq 0.20$. Improvement of the $T_g$ retrievals may be further explored with more complex modeling, auxiliary data, or a 2-parameter inversion. Previous studies have shown the effects of ground permittivity and snow density to L-band BTs at theoretical, tower-based radiometer, and satellite scales, Schwank et al. (2014); Lemmetyinen et al. (2016); Roy et al. (2017); Holmberg et al. (2024). We can expect the same for snow density and ground temperature. So a joined retrieval of $T_g$ and snow density may remove some artifacts due to the snow signal



in the retrieved $T_\mathrm{g}$ time series. However, additional prior information may have to be needed to ensure inversion stability. In the high-latitude areas, the revisit time is short. For all the sites, the median value of the difference between $T_\mathrm{g\text{-}insitu}$ at days $t$
and $t+1$ is 0.03°C. This difference remains at 0.1°C for a 3-day lag. Thus, $T_\mathrm{g\text{-}insitu}$ is very stable for short time range, which supports the thermal insulation of the snowpack. Considering a small temporal variation of $T_\mathrm{g}$ due to the snowpack thermal insulation, retrievals could be based on observations from multiple orbits (Konings et al., 2016). This could decrease the impact of the instrumental noise on the retrievals.

## 5.1 $T_\mathbf{g}$ retrievals under the snowpack for sites with $\nu_\mathbf{wi} \leq 0.04$

For sites with $\nu_\mathrm{wi} \leq 0.04$, correlation, bias and ubRMSD of the retrieval were superior to ERA5. A slightly negative bias was observed when the $\nu_\mathrm{wi}$ was ignored (using the model $\mathrm{MEM_G}$) but was successfully corrected with a model that accounts for snow and ice covered water bodies $\mathrm{MEM_{G+WI}}$.

### 5.1.1 Frozen ground parametrization

We used a frozen ground permittivity of $\varepsilon_\mathrm{frozen} = 5 + 0.5\,\mathrm{i}$, as defined by Hallikainen et al. (1985) and which was commonly
used in various studies (Schwank et al., 2014; Kerr et al., 2020; Holmberg et al., 2024). The emission depth of L-band observations is usually associated with the first 5 cm of the ground (Schmugge, 1983). However, the emission depth varies with the ground state and texture, based on the ground attenuation constant $\alpha$ ($\delta_e = 1/2\alpha$ Ulaby and Long (2014)), and consequently the ground complex dielectric constant $\varepsilon_\mathrm{g}$. For $\varepsilon_\mathrm{frozen} = 5.0 + 0.5\,\mathrm{i}$, the calculation based on Ulaby and Long (2014) shows that the associated emission depth $\simeq 15$ cm. When it comes to frozen ground, the effective depth is still not well defined and it
becomes even more complex with a snow layer on top of the ground. Rautiainen et al. (2012) estimated the emission depth of frozen ground at a maximum of 50 cm too, but observed a $T_\mathrm{B}$ saturation only when reaching a 30 cm frost depth. By computing metrics for $T_\mathrm{g\text{-}insitu}$ at all the available depths for the sites with $\nu_\mathrm{wi} \leq 0.04$, we found that R was better than ERA5 (median = 0.51) for depth down to 30 cm (median range from 0.57 to 0.74) (Figure 7). For *in situ* measurements down to 45 cm, the median absolute biases were smaller than 1.5°C and the median ubRMSD were smaller than 2.5°C. These results suggest that
the sensitivity depth is in fact down to 50 cm or less. For deeper $T_\mathrm{g\text{-}insitu}$, the correlation decreased to negative values (median R = -0.18 for depth = 120 cm). Note that for the period of this study (focused on $T_\mathrm{g\text{-}insitu}$ < -5°C at 5 cm depth) the ground was fully frozen down to 50 cm for the 11 USGS sites that provide ground temperatures down to 120 cm. Due to potential shallow frozen soil, emissions from the underlying unfrozen soil should be taken into account in the early winter (Schwank et al., 2004; Rautiainen et al., 2012).



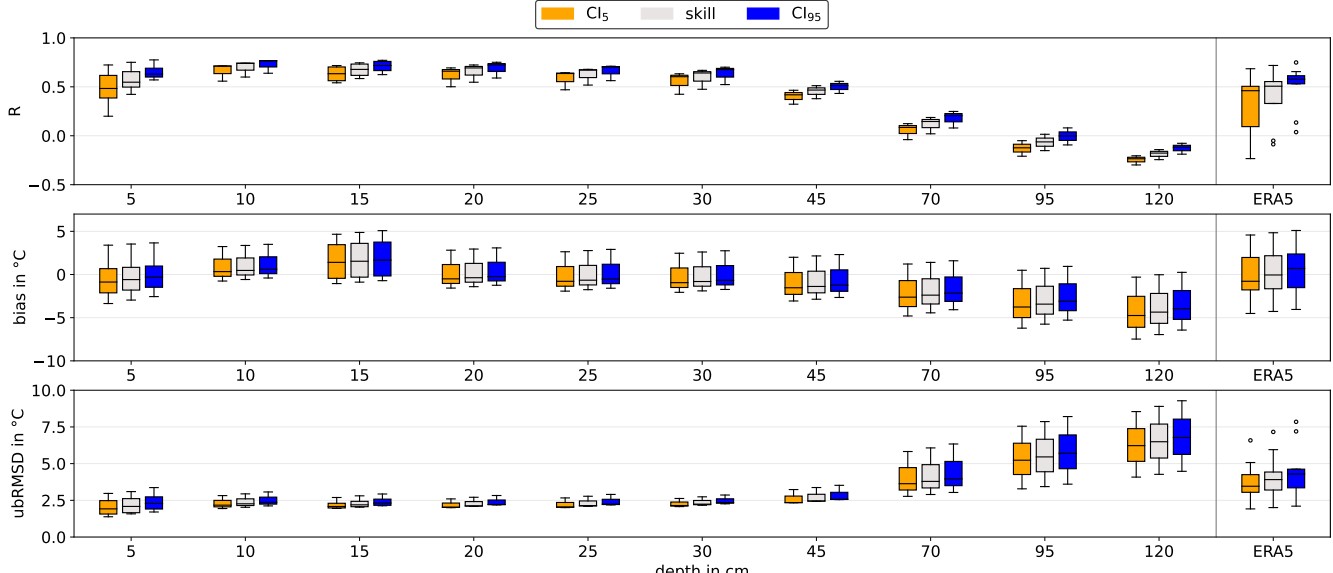

**Figure 7.** Summary statistics (in grey) of R, bias and ubRMSD and their 5% (in orange) and 95% (in blue) confidence intervals for sites with $\nu_{\text{wi}} \leq 0.04$. Boxes represent the site median and interquartile range ($Q_3 - Q_1$) and whiskers represent the 5 and 95 percentiles. The x-axis corresponds to the *in situ* probing depths used for the validation. The extreme right boxes are obtained with ERA5 and $T_{\text{g-insitu}}$ at 5 cm depth.

Concerning the ground surface parameters, the commonly used H-Q-N empirical model has been tuned for SM and VOD retrievals in many studies (Parrens et al., 2017; Chaubell et al., 2020; Preethi et al., 2024). Hence, its parametrization should be optimized for $T_{\text{g}}$ retrievals in arctic environment. We found the optimized set of values $H_{\text{r,gs}} = 0.8, Q_{\text{r,gs}} = 0, N^{\text{H}}_{\text{r,gs}} = 0, = N^{\text{V}}_{\text{r,gs}} = 0$ for the snow-ground interface, which is consistent with Holmberg et al. (2024). This parametrization depends on the chosen ground permittivity value. According to the Fresnel reflection coefficients (Equation 2), increasing ground permittivity

leads to a decrease of the emissivity. Using the H-Q-N model (Equation 3), increasing $H_{\text{r,gs}}$ means an increase of the emissivity. Thus, the soil parametrization requires a joint optimization of $\varepsilon_{\text{g}}$ and $H_{\text{r,gs}}$.

We optimized $H_{\text{r,gs}}$ based on a permittivity of a frozen ground value of $\varepsilon_{\text{frozen}} = 5 + 0.5$ i, but this value could be re-evaluated. The soil permittivity depends on the soil liquid water content and other characteristics (e.g. texture and bulk density). Based on a review of ground permittivity models (see Section 5.1.1), we investigated other potential values for frozen soil

permittivity. For a frozen ground ($T_{\text{g}} < -5°C$), we assumed the water to be completely frozen and thus SM negligible, i.e. $SM \simeq 0$ m$^3$ m$^{-3}$ (Zhang et al., 2010; Mavrovic et al., 2023). Soil property information (clay fraction, sand fraction, soil organic content, and bulk density) was extracted at each site location from the SoilGrids 250 m v2.0 database (Poggio et al., 2021) for the 0–5 cm soil layer (Table A1 in the appendices). The Soil Organic Carbon (SOC) content was very high at all the sites, as expected in the Arctic region, i.e five to ten times higher than the global mean 40 g kg$^{-1}$ (according to SoilGrid v2.0).

Dielectric constant models like the commonly used Mironov model do not use the SOC information to compute the permittivity.



It was first designed considering SM and clay content (Mironov et al., 2009). It was then further developed to use SM, $T_{\text{sg}}$ (here set as -20°C), and bulk density (Mironov et al., 2015). Park et al. (2017) was based on silt, clay, and sand contents, and bulk density. Bircher et al. (2016) defined a soil permittivity model tailored for high organic content soils, whereas Park's model was updated to consider soil organic content (Park et al., 2019). The permittivities computed with these models for our sites

are summarized in Table A2 in the appendices. The obtained $\varepsilon_{\text{frozen}}$ real parts went from 1 to 4, while the imaginary parts ranged from 0 to 0.1. This comparison of various permittivity models that depend on soil texture showed that the permittivity variability for frozen arctic soils was low and legitimate the use of a fixed value for the ground permittivity. However, the obtained permittivities were significantly lower than $\varepsilon_{\text{frozen}} = 5.0 + 0.5\,\text{i}$. This could be an evidence that $SM > 0 \text{ m}^3\text{ m}^{-3}$, even in frozen ground conditions ($T_{\text{g}} < -5°\text{C}$). *In situ* measurements of the frozen ground permittivity could be valuable,

simultaneously to tower-based radiometer observations in the Arctic tundra environment.

### 5.1.2 Effects of the snow layer

Snow cover was present for all ground temperature observations used in $T_{\text{g}}$ retrievals (i.e., the observed snow depth was above 10 cm), motivating the use of a snow layer in the MEM model. Lemmetyinen et al. (2016) and Roy et al. (2017) suggested that snow emissions at L-band are related to the bottom 10 cm of the snow layer. The typical Arctic snow profile consists of a

dense windslab of high density ($\rho \simeq 300 - 400 \text{ kg m}^{-3}$) but with a depth hoar underneath with lower density ($\rho \simeq 250 \text{ kg m}^{-3}$) (Sturm et al., 1997). However, the impact in terms of $\varepsilon_{\text{s}}$ is low in the model of Wiesmann and Mätzler (1999) that we used in the present study ($\varepsilon_{\text{s}}(\rho = 300 \text{ kg m}^{-3}) \simeq 1.5$ and $\varepsilon_{\text{s}}(\rho = 250 \text{ kg m}^{-3}) \simeq 1.4$). In addition, our model does not account for the inclusion of ice crusts in the snowpack (e.g. after rain on snow events) (Bartsch et al., 2023), nor low vegetation (e.g. shrubs or mosses) that could be observed in the tundra environment (Royer et al., 2021a) and might add complexity to the snowpack

microwave emission (Roy et al., 2018; Domine et al., 2022). Various temporal matching between *in situ* measurements $T_{\text{g-insitu}}$ and the retrieved $T_{\text{g}}$ were tested (not shown): closest measurement to the satellite overpass time (Catherinot et al., 2011) or daily maximum, minimum (Jones et al., 2007) or mean. The metrics remained similar because we observed very few daily variations of $T_{\text{g}}$ due to the snow insulation effect.

### 5.2 $T_{\text{g}}$ retrievals under the snowpack for sites with $\nu_{\text{wi}} \geq 0.20$

For sites with $0.20 \leq \nu_{\text{wi}} \leq 0.41$, the retrievals showed a strong negative bias when ignoring the snow and ice covered water bodies with $\text{MEM}_{\text{G}}$. We corrected the bias with the model $\text{MEM}_{\text{G+WI}}$ accounting for water bodies' contribution by optimizing the $H_{\text{r,wi}}$ parameter. A single $H_{\text{r,wi}}$ value did not suit all the sites. Validating $T_{\text{g}}$ retrievals for sites with water body fractions between 0.04 and 0.20 may help to understand the water bodies' effects in the retrievals and how to account for them. For sites with $\nu_{\text{wi}} \geq 0.44$, the bias was larger with $\text{MEM}_{\text{G+WI}}$ than with $\text{MEM}_{\text{G}}$. In fact, the bias could already be minimized using an

appropriate $H_{\text{r,gs}}$. However, the correlation remained poor for these sites (R < 0.3). For ERA5, the bias median was larger for sites with $\nu_{\text{wi}} \geq 0.20$ (median = 1.0°C) than for sites with $\nu_{\text{wi}} \leq 0.04$ (median = -0.1°C).





### 5.2.1 Effects of the snow and ice covered water bodies

We used the water fraction for a 40 km resolution, but Kerr et al. (2020) showed that a working area of $\sim 123\,km \times 123\,km$ is required to capture all the microwave signal that contributes to the SMOS observed BT. In fact, due to the multiple observation

angles, the size and shape of the elliptical footprint vary. Using an average single round buffer for all the angles is a potential error source. For sites located near the coast, the nearby presence of the ocean is non-negligible. The considered water body areas may also vary over time. Dynamic water maps could improve the $T_B$ correction, even more if they provide us with information on the water state (e.g. frozen, snow and ice covered, etc.). The water bodies highly impact the passive microwaves observations in summer in the Arctic area (Ortet et al., 2024). Including water bodies in the MEM in winter is even more

difficult because even if their surface is fully covered with ice, they may not be completely frozen in depth (Lemmetyinen et al., 2011). We tested various modeling configurations for the water bodies (ice only, liquid water only, ice on top of liquid water with a smooth interface, not shown). None were fully satisfying, but introducing the $H_{r,wi}$ parameter worked better. Indeed, it represents the surface roughness at the ice-water interface, which is not flat and significantly impacts microwave observations.

### 5.2.2 Analysis of a site with high water fraction (Inigok)

Figures 4 and 6 show that using a unique $H_{r,wi}$ for all the sites does not allow to get fully optimized $T_g$. To better understand the possible impact of snow and ice covered water bodies and model configuration, we present the Inigok site with a high water fraction of $\nu_{wi} = 0.23$. Figure 8 shows varying performance of the timeseries of $T_{g,MEM_G}$, $T_{g,MEM_{G+WI}}$ and $T_{g\text{-ERA5}}$ compared to $T_{g\text{-insitu}}$. The $T_{g,MEM_G}$ time series showed a negative bias that was well corrected in the $T_{g,MEM_{G+WI}}$ time series.

The $T_{g\text{-ERA5}}$ time series did not show a systematic bias with the $T_{g\text{-insitu}}$ time series. However, the ERA5 dynamic was quite different for the *in situ* measurements. While $T_{g\text{-insitu}}$ and $T_g$ seemed linked to air temperature when it rises above -10°C (e.g. in early 2014), but with a lag. This was not observed for $T_{g\text{-ERA5}}$, while it appeared in the retrieved $T_g$. This could be linked to wet snow events, that increase the snowpack conductivity and consequently the $T_g$. transparency. They also challenge the snowpack transparency hypothesis and could lead to an increase in the retrieved $T_g$ values. Using $MEM_G$ or $MEM_{G+WI}$

did not affect the time series dynamic, as shown by the similar R and ubRMSD in Figure 6. However, a strong interannual difference is observed. In winter 2014, we found R = 0.46 for $T_{g\text{-ERA5}}$, while we obtained R = 0.29 for both $T_{g,MEM_G}$ and $T_{g,MEM_{G+WI}}$ (see Figure B1 in the appendices). On the contrary, in winter 2019, a correlation of R = -0.03 is obtained with ERA5, while R = 0.61 using $MEM_G$ or $MEM_{G+WI}$. These discrepancies between years suggest that ice conditions change throughout the years and further ice parametrization would be needed to obtain satisfactory $T_g$ retrievals for scenes with high

water body fractions.



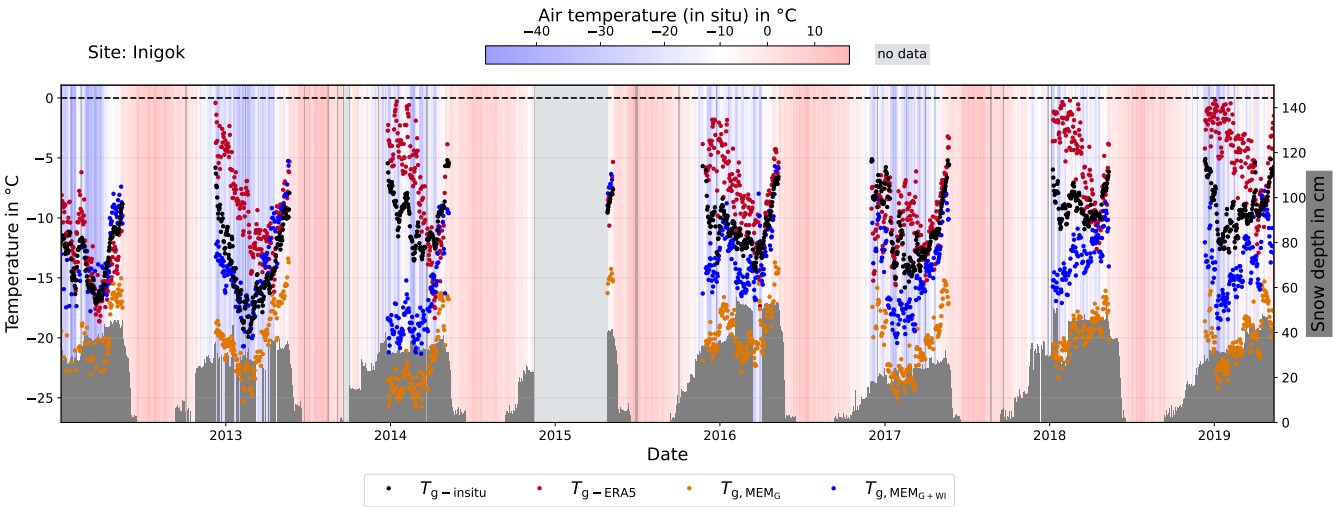

**Figure 8.** Time series of the ground temperatures (in °C) at Inigok from 2012 to 2020: $T_{\text{g-insitu}}$ (in black), $T_{\text{g,MEM}_\text{G}}$ (in orange), $T_{\text{g,MEM}_\text{G+WI}}$ (in blue) and $T_{\text{g-ERA5}}$ (in red). The snow depth (in cm) is displayed as dark grey bar plots. In the background, stripes from blue to red account for the *in situ* air temperature (in °C).

The similarities of behaviors of *in situ* and retrieved time series also varied during a single season. Figure 9 focuses on the retrievals using $\text{MEM}_\text{G+WI}$ with different $H_{\text{r,wi}}$ at Inigok. For each winter, the retrieved $T_\text{g}$ and $T_{\text{g-insitu}}$ were averaged per month and plotted with their standard deviation. Each graph of Figure 9 corresponds to a different $H_{\text{r,wi}}$ used in the modeling. December (mean = -0.3°C), January (mean = -0.4°C) and February (mean = 0.3°C) $T_\text{g}$ are in good agreement with $T_{\text{g-insitu}}$

for $H_{\text{r,wi}} = 0.7$. However, in March, $H_{\text{r,wi}} = 0.8$ provide better results (mean = -0.1°C). The best $H_{\text{r,wi}}$ is 0.9 for April (mean = 0.1°C) and May (mean = 0.3°C). This suggests a possible evolution of the ice conditions throughout the winter, that impacts the ice-water surface rugosity and $T_\text{g}$ inversion. This is in agreement with SAR studies (Duguay and Lafleur, 2003; Murfitt et al., 2023) which take into account roughness parameters over lakes to represent the impact of the roughness at the water-ice interface on microwave signal. Murfitt et al. (2023) linked the water-ice interface roughness with the growth of tubular bubbles

during ice formation, leading to higher roughness. Slushing water in ice cracks at the end of the freezing season induces more complexity than our three horizontal layers modeling for water bodies (Adams and Lasenby, 1985). Ground-based radiometric observations would be highly beneficial to better understand the seasonal effect of water-ice interface roughness on $T_\text{B}$ in Arctic regions.



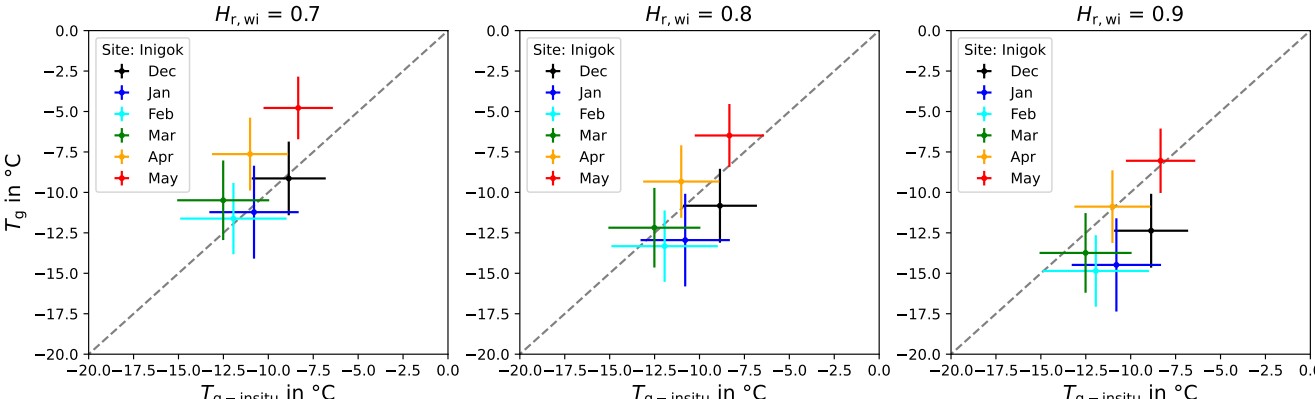

**Figure 9.** Scatter plots of the retrieved monthly average $T_g$ (in °C) against *in situ* averaged $T_{g\text{-insitu}}$ (in °C) at the Inigok site. The error bars show the standard deviation of the retrieved and measured temperatures. $H_{r,wi}$ values used in the inversion are 0.7 (left), 0.8 (middle), and 0.9 (right). The grey dashed line corresponds to the 1:1 identity line.

## 6 Conclusions

This study aimed to expand the previous studies on L-band passive microwave modeling and ground-based observations of snow-covered scenes by retrieving ground temperatures from satellite measurements in winter conditions. Our approach is based on SMOS L-band observations from 2012 to 2019. Two MEM configurations were explored to retrieve the $T_g$ below the snowpack in the Arctic: one considering a homogeneous scene ($MEM_G$) and another one correcting the scene for the snow and ice covered water body fraction ($MEM_{G+WI}$). $T_g$ retrieved with both MEM were validated with *in situ* measurements of 380 21 sites across northern Alaska and compared to $T_{g\text{-ERA5}}$. Several conclusions can be drawn from our results:

– $T_g$ under the snowpack can be retrieved from SMOS observations with a relatively simple MEM and limited auxiliary data.

– For sites with low water fraction ($\leq 0.04$), $T_g$ were retrieved with a median correlation R of 0.60 and a median bias of -0.2°C. For the same sites, the ERA5 median R was 0.51 and median bias was -0.8°C.

– For sites with a higher water fraction ($\geq 0.20$), ignoring the water fraction ($MEM_G$) leads to strong negative biases. The bias can be reduced using an ice-water roughness parameter $H_{r,wi}$, but correlation with *in situ* remains low ($< 0.5$ and worse than ERA5).

– Further work needs to be done to assess the impact of the snow and ice covered water bodies on L-band $T_B$ evolving through the winter season.



With its launch in 2010, SMOS has offered observations for almost 15 years to this day. Producing $T_\mathrm{g}$ maps over the Arctic for the whole period would improve monitoring of the permafrost state in space and time and would be highly beneficial for carbon models.

*Data availability.* SMOS L3BT are openly available at https://dx.doi.org/10.12770/6294e08c-baec-4282-a251-33fee22ec67f. USGS *in situ* data was sourced from https://www.sciencebase.gov/catalog/item/59d6a458e4b05fe04cc6b47e. CARVE data is freely available on https:
//daac.ornl.gov/cgi-bin/dsviewer.pl?ds_id=1424. SCAN and SNOTEL data was sourced from ISMN at https://ismn.earth/en/dataviewer/#. ERA5 data are openly available on https://cds.climate.copernicus.eu/datasets/reanalysis-era5-single-levels?tab=download. The ESA CCI L4 map, Version 2.0.7 can be accessed at http://maps.elie.ucl.ac.be/CCI/viewer/download.php.



## Appendix A: Soil properties

**Table A1.** Study sites soil characteristics at 0–5 cm extracted from SoilGrids 250 m v2.0 database (Poggio et al. 2021).

| Network | Site | Clay (%) | Sand (%) | Silt (%) | SOC (g kg$^{-1}$) | Bulk density (g cm$^{-3}$) |
|---|---|---|---|---|---|---|
| CARVE | Atqasuk | 14.1 | 67.2 | 18.7 | 402.3 | 0.33 |
| | Barrow | 28.3 | 37.8 | 33.9 | 360.7 | 0.51 |
| | Ivotuk | 25.4 | 29.3 | 45.3 | 384.3 | 0.43 |
| USGS | Inigok | 20.6 | 34.9 | 44.4 | 310.8 | 0.42 |
| | Fish Creek | 17.6 | 40 | 42.4 | 331.3 | 0.38 |
| | Umiat | 24 | 20 | 56 | 389.7 | 0.41 |
| | Tunalik | 20.3 | 31 | 48.7 | 331.3 | 0.45 |
| | Koluktak | 23.3 | 27.6 | 49.1 | 327.9 | 0.41 |
| | Niguanak | 19.8 | 31.8 | 48.3 | 279.3 | 0.47 |
| | Marsh Creek | 18.1 | 27.6 | 54.3 | 290.6 | 0.41 |
| | South Meade | 16.7 | 51.9 | 31.4 | 377.5 | 0.36 |
| | Camden Bay | 23 | 32.3 | 44.7 | 24.8 | 0.66 |
| | Awuna2 | 25.2 | 22.3 | 52.5 | 348.2 | 0.44 |
| | Piksiksak | 19.3 | 32.9 | 47.8 | 353.6 | 0.44 |
| | East Teshekpuk | 23.6 | 43.8 | 32.7 | 312.5 | 0.39 |
| | Ikpikpuk | 21.1 | 40.9 | 38.1 | 335.6 | 0.41 |
| ISMN SNOTEL | Imnaviat Creek | 16.7 | 41.6 | 41.7 | 337.2 | 0.35 |
| | Kelly Station | 14.5 | 30.2 | 55.3 | 286 | 0.55 |
| | Atigun Pass | 25 | 46 | 29 | 129.7 | 0.65 |
| ISMN SCAN | Ikalukrok Creek | 18.2 | 40.3 | 41.5 | 287 | 0.62 |



**Table A2.** Frozen soil permittivity $\varepsilon_{\text{frozen}}$ obtained from various dielectric constant models, with SM $= 0$ m$^3$ m$^{-3}$ and other soil properties from SoilGrid 250 m v2.0 (Poggio et al., 2021) (Table A1). Note that the sign before the imagery part depends on different conventions.

| Network | Site | Mironov et al. (2009) | Mironov et al. (2015) | Park et al. (2017) | Park et al. (2019) |
|---|---|---|---|---|---|
| CARVE | Atqasuk | 2.36 + 0.11 i | 1.45 + 0.04 i | 2.22 + 0.07 i | 1.91 + 0.06 i |
| | Barrow | 2.15 + 0.08 i | 1.73 + 0.06 i | 2.07 + 0.07 i | 2.17 + 0.08 i |
| | Ivotuk | 2.19 + 0.09 i | 1.60 + 0.05 i | 2.36 + 0.09 i | 2.23 + 0.09 i |
| USGS | Inigok | 2.26 + 0.10 i | 1.59 + 0.05 i | 2.33 + 0.09 i | 2.18 + 0.09 i |
| | Fish Creek | 2.30 + 0.10 i | 1.53 + 0.04 i | 2.39 + 0.10 i | 2.13 + 0.08 i |
| | Umiat | 2.21 + 0.09 i | 1.57 + 0.05 i | 2.50 + 0.11 i | 2.31 + 0.10 i |
| | Tunalik | 2.26 + 0.10 i | 1.64 + 0.05 i | 2.29 + 0.09 i | 2.21 + 0.09 i |
| | Koluktak | 2.22 + 0.09 i | 1.57 + 0.05 i | 2.43 + 0.10 i | 2.25 + 0.09 i |
| | Niguanak | 2.27 + 0.10 i | 1.67 + 0.06 i | 2.22 + 0.09 i | 2.21 + 0.09 i |
| | Marsh Creek | 2.30 + 0.10 i | 1.57 + 0.05 i | 2.43 + 0.11 i | 2.23 + 0.09 i |
| | South Meade | 2.32 + 0.10 i | 1.50 + 0.04 i | 2.32 + 0.09 i | 2.04 + 0.07 i |
| | Camden Bay | 2.22 + 0.09 i | 1.98 + 0.09 i | 1.71 + 0.06 i | 2.24 + 0.09 i |
| | Awuna2 | 2.19 + 0.09 i | 1.62 + 0.05 i | 2.39 + 0.10 i | 2.29 + 0.10 i |
| | Piksiksak | 2.28 + 0.10 i | 1.62 + 0.05 i | 2.30 + 0.09 i | 2.19 + 0.09 i |
| | East Teshekpuk | 2.21 + 0.09 i | 1.54 + 0.05 i | 2.33 + 0.09 i | 2.12 + 0.07 i |
| | Ikpikpuk | 2.25 + 0.09 i | 1.57 + 0.05 i | 2.30 + 0.09 i | 2.14 + 0.08 i |
| | Lake 145 | 2.23 + 0.09 i | 1.72 + 0.06 i | 2.05 + 0.07 i | 2.12 + 0.08 i |
| ISMN SNOTEL | Imnaviat Creek | 2.32 + 0.10 i | 1.48 + 0.04 i | 2.45 + 0.10 i | 2.12 + 0.08 i |
| | Kelly Station | 2.36 + 0.10 i | 1.80 + 0.07 i | 2.02 + 0.08 i | 2.20 + 0.09 i |
| | Atigun Pass | 2.19 + 0.09 i | 1.97 + 0.09 i | 1.66 + 0.05 i | 2.12 + 0.07 i |
| ISMN SCAN | Ikalukrok Creek | 2.29 + 0.10 i | 1.92 + 0.08 i | 1.77 + 0.06 i | 2.14 + 0.08 i |



## Appendix B: Case study: Inigok

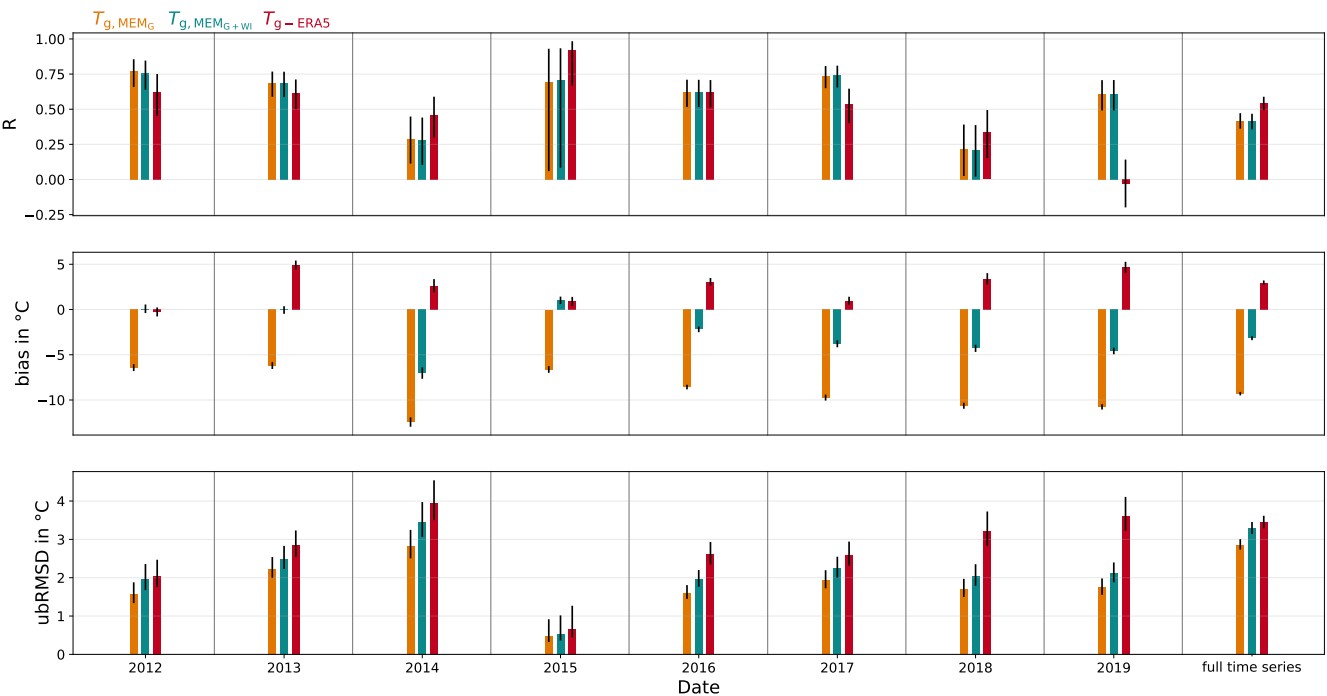

**Figure B1.** Yearly metrics obtained at Inigok from 2012 to 2020: $T_{\mathrm{g,MEM_G}}$ (in orange), $T_{\mathrm{g,MEM_{G+WI}}}$ (in blue) and $T_{\mathrm{g\text{-}ERA5}}$ (in red). R, bias and ubRMSD are plotted as bar plots, with error bars accounting for their 5% and 95% confidence intervals. On the far right, we show the global metrics obtained for the whole timeseries.

*Author contributions.* JO carried out this study by analyzing data, performing the inversions, and organizing and writing the paper. ARoyer, AM and ARoy proposed the initial idea. MS and MH provided expertise in microwave emission model and contributed to the writing of the manuscript. All the authors were involved in the analysis of the results and contributed to the writing of the paper.

*Competing interests.* The authors declare no conflict of interest.

*Acknowledgements.* This work was funded by the CNES (Centre National d'Etudes Spatiales) through J.O. PhD funding (contract no.
JC.2O2O.OO39O41) and the Science TOSCA (Terre Océan Surfaces Continentales et Atmosphère) program. The authors acknowledge the support of the Natural Sciences and Engineering Research Council of Canada (NSERC). This study has been partially supported through the grant EUR TESS N°ANR-18-EURE-0018 in the framework of the Programme des Investissements d'Avenir. A contribution to this work



was made at the Jet Propulsion Laboratory, California Institute of Technology, under a contract with the National Aeronautics and Space Administration.



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
