# Peer review of "Retrieving frozen ground surface temperature under the snowpack in Arctic permafrost area from SMOS observations"

_EGUsphere, 2024_

## Author Response (AR1)

*Retrieving frozen ground surface temperature under the snowpack in Arctic permafrost area from SMOS observations*

**Comments from Reviewer 1**

Comments from the reviewer
Answers from the authors

**Reviewer 1: General Comments**

In this study the authors developed an approach for estimating frozen ground surface temperature underlying snow in Arctic tundra environments. Their approach is based on the inversion of a relatively simple microwave emission model (MEM), tailored to arctic tundra winter conditions, and driven by SMOS L-band brightness temperature observations. Differing MEM types are applied; the first assuming homogeneous surface conditions and the second applying a fractional water cover (FW) correction for snow and ice covered water bodies to reduce associated bias on ground temperature retrievals. The retrievals were derived and evaluated against global reanalysis data (ERA-5) and in situ temperature measurements across 21 Arctic tundra reference sites in northern Alaska permafrost locations from 2011-2020. The results reveal the important influence of FW on SMOS ground temperature retrievals in the Arctic and demonstrate an effective method for temperature retrievals spanning a range of FW conditions characteristic of the Arctic tundra winter environment.
Overall, the paper is well written with clearly explained methods and interesting and well justified results, and conclusions. The figures and table summaries are clearly depicted and provide adequate support to the major points of the paper. The study also shows the potential for effective monitoring of Arctic winter ground temperatures beneath the tundra snowpack from SMOS and other low frequency satellite microwave radiometers using a relatively simple MEM with minimal ancillary data requirements. Significant broader impacts from this study include the potential for satellite based monitoring of winter soil temperatures in the Arctic, which are generally poorly characterized in global land models and from available sparse monitoring sites. Winter ground temperatures also have strong science value due to amplified Arctic warming trends and the strong association of ground temperatures on permafrost and soil carbon stability. I consider the paper suitable for publication in it's present form pending consideration of the following minor revisions.

*Reply* : Many thanks for your valuable comments. Please find our responses below. We have numbered the items for referencing. We are limiting the response to items highlighting potential issues and stating specific questions and we corrected all suggested editorial changes.

**Reviewer 1:**

**R1-C1**

Section 1: Include a concise statement of the broader science objective or goal of the study at the end of the Introduction section.

*Reply* : A sentence was added at the end of to the Introduction, highlighting the the broader science objective explore in this study.

*New* (56 - 57): "This satellite-based approach opens a new path towards soil temperatures monitoring under the snowpack in the Arctic with expected improvement in land and carbon cycle modeling in permafrost area."

**R1-C2**

Section 2.4: Given the enhanced influence of topography on microclimate heterogeneity at high latitudes, consider adding the terrain elevation heterogeneity surrounding site location grid as an additional factor that may help explain differences between the relatively coarse resolution satellite and reanalysis observations, and the in situ site measurements.

*Reply* : Topography can indeed impact the SMOS measured brightness temperatures by up to 5 K (Kerr et al., 2004). However, one has to keep in mind that the topography at L-band and at SMOS scale is different from the topography for optical sensors. Mialon et al. (2008) identified SMOS pixels affected by the topography based on an emissivity model and Digital Elevation Model (DEM, from the Shutter Radar Topography Mission

-STRM). They evaluated the impact of topography on the SM retrieval, and classified SMOS footprints as: not impacted by topography, slightly impacted by topography, i.e. SMOS SM should be considered with care; significantly impacted by topography, i.e. SMOS data should not be used. We carefully checked our sites, and none of the SMOS footprint associated with the 21 study sites is affected by strong topography and only Atigun Pass is classified with few "moderate" topography. Precisions about topography checking were added in Section 2.2.

*Old* (l. 69 - 72): "The 21 reference in situ sites are located across Alaska (US), in the Arctic region (Figure 1 and Table 1). The topography is flat and the continuous permafrost landscape integrates numerous lakes. Some sites are located close to the coast (Barrow, Lake 145, Fish Creek, Camden Bay) while others are disseminated inland. All the selected sites are located above the tree line and are representative of the tundra environment with vegetation characterized by low shrubs and mosses (Table 1)."

*New* (71 - 76): "The 21 reference in situ sites are located across Alaska (US), in the Arctic region (Figure 1 and Table 1). The continuous permafrost landscape integrates numerous lakes and some sites are located close to the coast (Barrow, Lake 145, Fish Creek, Camden Bay) while others are disseminated inland. All the selected sites are located above the tree line and are representative of the tundra environment with vegetation characterized by low shrubs and mosses (Table 1). *SMOS observations are flagged for topography (Mialon et al., 2008), but none of the 21 in situ sites are impacted, except for the Atigun pass site which is labeled as moderate topography, i.e. SMOS data quality may be impacted by topography.*"

**R1-C3**

Section 4.2.2: Do the sites with higher bias share similar features that may help account for the larger temperature error? E.g., sites located along coastlines near open ocean or in complex foothills topography may be expected to have larger apparent bias than relatively flat inland locations.

*Reply* : We indeed expected a dependency with surface characteristics. This motivated Table 1 and Table A2 of our manuscript. Unfortunately, it was difficult to conclude on the influence of surface on the performances of the retrieval. Neither the land cover (Table 1), nor the soil organic and clay content (Table A2) gave a significant result. For sites located by the coast, the Artic sea water/ice may have a different impact than inland waters. But, mostly, Gutierrez et al. (2012) showed that the existence of land-sea contamination effects which can be mitigated in the image reconstruction. We added some of these inputs in the discussion section:

*New* (359 - 364): "For sites with higher biases (namely Niguanak, Marsh Creek, Camden Bay and Fish Creek), no correlation could be made with surface characteristics, such as land cover (Table 1) and soil content (Table A2). However, we noticed that those sites correspond to coastal pixels, i.e. made of BT measured on the continent and the ocean. Kerr et al. (2020); Gutierrez et al. (2012) highlighted the retrieval difficulties for coastal BT that result from mixed pixels of land and sea. In fact, the observation geometry variations that lead to various water fractions are not taken into account in the MEM and difficult to model."

**Retrieving frozen ground surface temperature under the snowpack in Arctic permafrost area from SMOS observations**

**Comments from Reviewer 2 / Christian Matzler**

Comments from the reviewer
Answers from the authors

**Reviewer 2: General Comments**

This work is an effort to estimate ground temperature (usually below a snowpack) in arctic tundra regions from SMOS data, using for validation in-situ data in Alaska at fixed stations (latitude range from 68° to 71°N). Comparison is also made with ERA 5 data.
The retrieval model is simple, using only two fitting parameters Hr , one for the snow-ground interface and the other one for the ice-water interface. Unfortunately, the model test only relies on statistics, assuming constant behaviour for the investigated 8-year period.
A problem is the strong and variable influence of water bodies on the SMOS data. Furthermore, the large footprint of the satellite data limits the representativeness of local station data. Nevertheless, in areas with small water fraction, the results show promising results. Further work is needed because wetlands, rivers and lakes with variable snow and ice cover are abundant in this area. Their influence on microwave emission cannot be accounted for with the present assumptions, such as constant H r values. Solutions may have to use further information, e.g. from polarisation, see references below.

*Reply* : We acknowledge the reviewer for the comments, and for suggesting thoughtful literature on the topic. The remarks are indeed justified, and this study is a first step to demonstrate the faisability of the approach. We develop some points raised by the reviewer in our responses below.

**Reviewer 2:**

**R2-C1**

Fitting parameters: The model used is very simple. Although the Hr values are thought to be related to interface roughness, other effects, such as impedance matching and local absorption/emission also play a role. Reduced reflectivity (Equation (3)) means increased absorption/emission. Therefore, such effects can be simulated using this model, too. An example is a sudden inflow of liquid water into the interface layers, e.g. by wind braking ice. But this means that the parameters are variable in space and time.

*Reply* : The model proposed in this study is, in fact, simple. Various choices led us to assess this method that is applicable at the global scale. First, very few auxiliary data are required in the inversion process. It is only based on the SMOS observations (12 years of L3 BT), modeled temperatures from ECMWF and landcover characterization from ESA CCI. Consequently, the retrievals can be made at global scale, i.e. in the whole circumarctic permafrost area.

In addition, the inner heterogeneity of the SMOS field of view and its consequences are still difficult to assess (Gibon et al., 2022, 2024). Adding some auxiliary data to the process would require a detailed study of the additional information at the SMOS scale. Similarly, setting temporal varying parameters in the model would lead to increasing the complexity.

Few studies explored the complex relations that link transmissivity, reflectivity, absorption, and emission theoretically (Mätzler, 2006; Schwank et al., 2015; Naderpour et al., 2017b) or at the local scale (Mätzler, 1994; Naderpour et al., 2017a). We agree that the $H_\mathrm{r}$ parameter is not only a matter of interface geometrical roughness, but also accounts for local heterogeneity and has various impacts on parameter retrievals (Naderpour et al., 2017a; Holmberg et al., 2024). Yet, it still appears to be useful for simple modeling optimization and is considered as an empirical fitting parameter as in Lemmetyinen et al. (2011).

We recognized that the parameterization of lake ice is over simplified. Indeed, as Figure 8 and 9 show, it seems that ice conditions change in time. This was observed by many studies (Adams and Lasenby, 1985; Duguay and Lafleur, 2003; Murfitt et al., 2023). However, very few studies have tried to simulate the various processes affecting lake ice L-Band signal. Our study underlines the importance to improve that understanding in order to improve our algorithm in the future. We added few sentences in the discussion to clarify that point:

*New* (409 - 410): "Such observations may also help the development of a more complex model to better describe the L-band emissions of the circumarctic lakes and their variations through the seasons."

**R2-C2**

A more general model may have to consider additional effects such as:

- The reflection at the boundary between frozen and unfrozen soil may have a contribution to the observed signal.
  *Reply* : Our first tests were to considered a soil with a water content and perform a joint SM and $T_g$ retrieval. With the SMOS observations, it led to very low SM content. Based on the in situ temperatures (see Figure 1), that show frozen conditions down to 120 cm, and considering the estimated emission depth ( $\delta_e \simeq 15$ cm as suggested by Ulaby and Long (2014) or $\delta_e \simeq 50$ cm, as suggested by Rautiainen et al. (2012)), we concluded that as a first order hypothesis, the deeper unfrozen soil did not impact much our retrieval. Note that the hypothesis is valid in the context of our modeling approach.

[Figure]

Figure 1: Boxplots of the *in situ* ground temperatures (in °C) at various depths for all sites. Temperatures are selected from 2012 to 2020 when inversion is performed, i.e. $T_{g\text{-insitu}} < -5$°C at the shallower depth available (5 cm or 15 cm). The boxes represent the site median and interquartile range ($Q_3 - Q_1$), the whiskers represent the 5 and 95 percentiles and the points are outliers.

*New* (299 - 300): "As observed by Schwank et al. (2004), the observed signal can encompass a contribution of the boundary between frozen and unfrozen soil, which was not taken into account in our modeling."

- In permafrost areas, the soil layer may freeze completely.
  *Reply*: Indeed, and this was confirmed by *in situ* measurements down to 1 m depth for our sites, as shown in Figure 1. It shows the box plot of the temperatures per site, and per depth during the studied period (i.e. winter). The deepest measurements (120 cm) show average temperatures around -10°C and, in our selected period, 99% of the measurements at all depths are below -1.6°C.

- Shallow lakes may freeze down to the bottom.
  *Reply* : This is an interesting point. We checked in the literature about the state of lakes in winter. No reliable database exists on the topic. We found information on the surface extent from MODIS/Landsat and on the yearly extend from :

  - Pekel et al. (2016): however this only considers the unfrozen period,
  - Klein et al. (2024) (Global WaterPack): but areas above 70°N are missing.

Importantly, we tried to evaluate the impact of lakes (as frozen and unfrozen) with our model. Considering frozen water, we were not able to fit SMOS observations, and we had to consider open water in our model to significantly decrease the modeled BT to fit the SMOS BT (see R2-C1).
*New* (377 - 378): "Yet, different models should be applied depending on water bodies characteristics (e.g. depth) as shallower lakes could freeze down to bottom or sea ice may be formed on the coastal areas."

- Bare rock areas with dielectric constant different from soil.

  *Reply*: Indeed. We checked the presence of barren soil in the land cover classification, and found that only 2 of our study sites were concerned with barren rock, i.e. Atigun Pass (24%) and Ikalukrockcreek (11%) above 10%.

  Barren soil is mostly composed with carbonate (Martha Raynolds, 2022), and the corresponding dielectric constant is estimated to $\varepsilon_{\text{carbonate}} \simeq 8 + 0.2$ i (Table 5.8. of Mätzler (2006)). We performed a sensitivity analysis to evaluate the impact of using a $\varepsilon_g = 5 + 0.5$ i instead of $\varepsilon_{\text{carbonate}} \simeq 8 + 0.2$ i. It resulted in a $\Delta T_{g,\text{inv}} - T_{g,\text{obs}} \simeq -8.1^oC$ (see R3-C4). The impact of using another dielectric constant is not deniable but it would require a precise BDD of all soil types and the use of dielectric mixing formula (see R2-C3).

- Influence of vegetation may be noticeable.

  *Reply*: This is actually a study we are performing as this topic seems of interest. We first wanted to evaluate the possibility of deriving soil temperature during winter conditions (this manuscript). Our current work evaluates the impact of vegetation within the snowpack on the derived $T_g$. We aim to assess the difficulties and impacts of a snowpack with vegetation with various sensitivity analysis. In fact, the effects of vegetation can be observed in the snow density and variations in snow density affect the $T_g$ retrieval (see R3-C5). We added some precision about vegetation effects and modeling in the discussion :

  *New* (344 - 349): "As for the vegetation, multiple effects may mitigate the $T_g$. The presence of shrubs leads to a snow accumulation with a lower density than on herbaceous areas, which means more thermal insulation from the snowpack (Grünberg et al., 2020; Liston et al., 2002). However, Domine et al. (2022) also observed thermal exchanges between air and soil through the branches. As these effects are observed at local scale, it is difficult to model it at the SMOS scale ($\simeq 40$ km)."

- Rain-on-snow events followed by refreezing.

  *Reply*: Such events impact significantly the emissivity of microwave, and is currently studied at 19 and 37 GHz (Grenfell and Putkonen, 2008). Moreover, Dolant et al. (2018) makes an inventory of ROS events. Experiments are also conducted in Cambridge Bay with an L-band radiometer (as part of a Sherbrooke University PhD), but no analysis were made available. This may be to considered in future analysis, and possible improvements of our methods. We would need to first detect such events at SMOS scale ($\sim 40$ km).

  A sentence was added to the discussion :

  *New* (343 - 344): "In fact, Roy et al. (2015) observed a decrease in horizontal polarization as the impact of ice crust formation, but Roy et al. (2018) underlined the difficulty of modeling and quantifying such event at L-band."

Some of these effects may be identified by temporal variations, giving valuable information, as can be seen in Figure 8.

**R2-C3**

The value, 5.0, for the real part, needs a clarification, that may explain the discrepancy to the smaller values in Table A2: When soil freezes in late fall, the soil is often water saturated at and near the surface due to dew formation and water-vapour migration from warmer soil below. The value of 5 represents the dielectric constant of the frozen version of this kind of soil. This is also the reason for the large contrast of microwave signatures between frozen and unfrozen soil reported by many observers. In arctic regions, especially in high-porosity organic soil, the situation may be different. The imaginary part is sensitive to soil type, but values decrease with decreasing temperature.

*Reply* : Thank you for theses valuable precisions about the frozen soil dielectric constant. The choice of the frozen ground dielectric constant has been discussed in :

*New* (325 - 333): "In addition, a permittivity equal to $\varepsilon_{\text{frozen}} = 5.0 + 0.5$ i may result from a soil surface which was saturated with water at freezing time. But, as the Arctic soil shows high SOC and high bulk density (Table A1), it may not satisfy this water saturation condition. For the imaginary part of the permittivity, Mironov et al. (2015) showed a decrease with decreasing temperatures. *In situ* measurements of frozen ground permittivity could be valuable, simultaneously to tower-based radiometer observations in the Arctic tundra environment. Some probes seem efficient for this task, such as the one described in Gélinas et al. (2025).Using a constant permittivity, calculated under the assumption of a homogeneous ground, is a practical solution for our model, as it reduces the number of free parameters and auxiliary data. However, dielectric mixing models enable to characterize heterogeneous materials (Ulaby and Long, 2014) and could better fit the local behavior of Arctic soils."

**R2-C4**

I do not fully understand the data in yellow, grey and blue. All three of them essentially show the same. The figures should be simplified. Some extra points are unexplained. Furthermore, in Figure 5, and in its caption, there appear to be errors with regard to Hr. The remark about the x axis is unclear and confusing. I have the same problem with Figure B1.

*Reply* : Indeed, Figures 5/6/7 displayed a lot of information. We simplified these figures as recommended (Figures 2, 3, 4). However, we consider that the confidence intervals are useful for data analysis and interpretation, so we kept this information as an appendix (see Appendix B: Results: Figures with confidence intervals). We also corrected the caption of Figure 5 such as:

*New*: "Summary statistics of R, bias and ubRMSD for sites with $\nu_{\mathrm{wi}} \leq 0.04$. The boxes show the median and interquartile range and whiskers show the 5 and 95 percentiles obtained from all the considered sites. The boxes correspond to the skill estimate (R, bias, or ubRMSD). The associated 5% and 95% CI are provided in Figure B1. The x-axis corresponds to the $H_{\mathrm{r,wi}}$ used in the inversion. The boxes are respectively obtained from: $\mathrm{MEM_G}$ with $H_{\mathrm{r,gs}} = 0.8$ (left), $\mathrm{MEM_{G+WI}}$ with $H_{\mathrm{r,wi}} = 1$ (center) and ERA5 (right)."

[Figure]

Figure 2: Summary statistics of R, bias and ubRMSD for sites with $\nu_{\mathrm{wi}} \leq 0.04$. The boxes show the median and interquartile range and whiskers show the 5 and 95 percentiles obtained from all the considered sites. The boxes correspond to the skill estimate (R, bias, or ubRMSD). The associated 5% and 95% CI are provided in Figure B1. The x-axis corresponds to the $H_{\mathrm{r,wi}}$ used in the inversion. The boxes are respectively obtained from: $\mathrm{MEM_G}$ with $H_{\mathrm{r,gs}} = 0.8$ (left), $\mathrm{MEM_{G+WI}}$ with $H_{\mathrm{r,wi}} = 1$ (center) and ERA5 (right).

[Figure]

Figure 3: Summary statistics of R, bias and ubRMSD for sites with $0.20 \leq \nu_{\mathrm{wi}} \leq 0.41$. The associated 5% and 95% CI are provided in Figure B2. Boxes represent the site median and interquartile range ($Q_3$ - $Q_1$) and whiskers represent the 5 and 95 percentiles. The x-axis corresponds to the $H_{\mathrm{r,wi}}$ used in the inversion. The rightmost boxes are obtained with ERA5.

[Figure]

Figure 4: Summary statistics of R, bias and ubRMSD for sites with $\nu_{\mathrm{wi}} \leq 0.04$. The associated 5% and 95% CI are provided in Figure B3. Boxes represent the site median and interquartile range ($Q_3$ - $Q_1$) and whiskers represent the 5 and 95 percentiles. The x-axis corresponds to the *in situ* probing depths used for the validation. The extreme right boxes are obtained with ERA5 and $T_{\mathrm{g\text{-}insitu}}$ at 5 cm depth.

**R2-C5**

For me this ist the most interesting figure of the study. It shows temporal variations that support support the applied retrieval model, e.g. in 2012 and 2013, but with significant differences in other years. To understand the behaviour, the data should be compared with additional in-situ information and with meteorological data. This may be helpful for the understanding, and consequently for the refinement of the retrieval model.

*Reply* : We agree that additional in situ data would be helpful to better understand the signal. However, all the available in situ data are already displayed in the Figure 8, as: ground temperature, air temperature and snow depth. It has to be mentioned that data are very sparse in these remote and harsh environments. However, we modified the Figure 8 to make it clearer, especially the legend to better emphasize the use of in situ data. The whole legend is now below the figure, and the caption was modified to indicate the axis meaning.

[Figure]

Figure 5: Time series of the ground temperatures (in °C, left axis) at Inigok from 2012 to 2020: $T_{\text{g-insitu}}$ (in black), $T_{\text{g,MEM}_{\text{G}}}$ (in orange), $T_{\text{g,MEM}_{\text{G+WI}}}$ (in blue) and $T_{\text{g-ERA5}}$ (in red). The snow depth (in cm, right axis) is displayed as dark grey bar plots. In the background, stripes from blue to red account for the *in situ* air temperature (in °C).

**R2-C6**

- Line 9: Please define: median correlation R

  *Reply* : We removed the term correlation to clarify the expression as "median R".

  *New* (9): "For sites with water fraction < 0.04, our methods (median R = 0.60) outperformed the European Centre for Medium-Range Weather Forecasts reanalysis (ERA5) product (median R = 0.51) with respect to the reference sites."

- Line 353: Need for clarification: snowpack conductivity: thermal or electrical?

  Same line: T g transparency, what do you mean? Please define or explain.

  *Reply* : We refered here to the snowpack thermal conductivity. The use of "transparency" was a typo. We precised the sentences :

  *New* (386 - 389): "This was not observed for $T_{\text{g-ERA5}}$, while it appeared in the retrieved $T_{\text{g,MEM}_{\text{G}}}$ and $T_{\text{g,MEM}_{\text{G+WI}}}$. This could be linked to wet snow events, that increase the snowpack thermal conductivity and consequently the link between air temperatures and $T_{\text{g}}$. They also challenge the snowpack transparency hypothesis (Kumawat et al., 2022), that could be not valid anymore, and could lead to an increase in the retrieved $T_{\text{g}}$ values."

- Lines 364 to 366: What do the temperature values in brackets () mean?

  *Reply* : We defined the meaning of the values in brackets as :

  *New* (399): "The difference between the monthly averaged $T_{\text{g}}$ and the monthly averaged $T_{\text{g-insitu}}$ is noted $\Delta\bar{T}$. December ($\Delta\bar{T}$ = -0.3°C), January ($\Delta\bar{T}$ = -0.4°C) and February ($\Delta\bar{T}$ = 0.3°C) $T_{\text{g}}$ are in good agreement with $T_{\text{g-insitu}}$ for $H_{\text{r,wi}}$ = 0.7. However, in March, $H_{\text{r,wi}}$ = 0.8 provide better results ($\Delta\bar{T}$ = -0.1°C). The best $H_{\text{r,wi}}$ is 0.9 for April ($\Delta\bar{T}$ = 0.1°C) and May ($\Delta\bar{T}$ = 0.3°C). "

- Figure 9: Based on the variable behaviour from year to year, the figure should focus on single years, first.

  *Reply* : The figure 9 now only focuses on year 2017.

[Figure]

Figure 6: Scatter plots of the retrieved monthly average $T_\mathrm{g}$ (in °C) against *in situ* averaged $T_\mathrm{g\text{-}insitu}$ (in °C) at the Inigok site from December 2016 to May 2017. The error bars show the standard deviation of the retrieved and measured temperatures. $H_\mathrm{r,wi}$ values used in the inversion are 0.7 (left), 0.8 (middle), and 0.9 (right). The grey dashed line corresponds to the 1:1 identity line.

- Table A2: Clarify SM in the caption. Frozen or unfrozen water content? Please give information on both.
  *Reply* : Soil moisture definition for frozen soil is complex. Here we consider the unfrozen water SM = 0 m³ m⁻³. The soil permittivity is considered to be calculated from a homogenous soil, i.e. pure soil without frozen water (see R2-C3).
  This was clarified in:
  *New*: No unfrozen water is considered, i.e. SM = 0 m³ m⁻³.

*Retrieving frozen ground surface temperature under the snowpack in Arctic permafrost area from SMOS observations*

**Comments from Reviewer 3**

Comments from the reviewer
Answers from the authors

**Reviewer 3: General Comments**

This paper presents a novel method for retrieving ground surface temperatures under snowpack in Arctic permafrost regions using Soil Moisture and Ocean Salinity L-band brightness temperature observations. The study is well-motivated and addresses a critical gap in monitoring Arctic permafrost dynamics. The authors develop and evaluate two microwave emission models to retrieve ground surface temperatures and validate their results against in situ measurements and ERA5 reanalysis data.
The use of two models shows a thoughtful approach to handling complex Arctic environments. Additionally, the optimisation of surface roughness parameters is a key strength in their methodology as it improves the accuracy of retrievals, especially in areas with significant water fractions. The decision to validate against 21 sites provides an effective evaluation of the method's performance. The paper is well-organized, and generally, figures and tables are well-designed and effectively support the text.
Overall, the methodology is sound, and the results are promising, particularly for regions with low water fractions. However, the paper would benefit from a more detailed discussion of its limitations, broader implications, and uncertainties. With some revisions, this paper will significantly contribute to the field of remote sensing and cryosphere studies. I will recommend this paper be accepted for publication after addressing the major and minor revisions outlined below.

We are thankful for the positive review of our article. We appreciate the valuable remarks and references to enrich the manuscript.

**Reviewer 3**

**R3-C1**

The discussion section could be expanded to address the broader implications of the study for Arctic climate research and operational monitoring. This would enhance the paper's impact and relevance to a broader community.

*Reply* : We added materials about follow-on objectives and implications for the Arctic climate and carbon cycle research.

*New* (260 - 266): "In addition, this satellite-based approach is a first attempt to monitor the soil temperatures under the snowpack in the whole circumarctic permafrost area. Based on L-band observations of SMOS since 2010, continuing efforts in long-term and operational permafrost state monitoring would be made possible by the upcoming satellite missions CIMR and CryoRad (Donlon et al., 2023; Macelloni et al., 2018). Such soil temperature measurements would be highly beneficial for climate monitoring and carbon cycle modeling. Future work will look at integrating our approach to assimilation approaches such as the SMAP L4 (Jones et al., 2017) to improve soil temperature in winter and winter soil $CO_2$ emission."

**R3-C2**

The limitations of the method, particularly for sites with high water fractions, should be addressed more thoroughly. The authors could propose specific strategies for improving the model in these regions.

*Reply* : Many persisting difficulties remain about frozen open water area modeling in passive microwaves. Based on Pekel et al. (2016); Klein et al. (2024), we made sure that the water bodies do not change in terms of yearly extent. Yet we are missing a precise map of the extent during winter time and of the state of open water during this period (completely frozen or unfrozen with a ice layer on top). One possibility would be to consider high-resolution data from the SWOT mission (Biancamaria et al., 2016). In addition, the SMOS derived data could be disaggregated to obtain better resolution products (Molero et al., 2016). Yet, modeling

the emissions of lakes throughout winter remains a heavy task (see R2-C1, R2-C2). The development of high resolution missions following SMOS/SMAP is crucial but current best strategy is to get rid of open water on the field of view (Rodriguez-Fernandez et al., 2022, 2024).

**R3-C3**

Would it be possible to introduce site-specific roughness optimization or incorporate additional auxiliary datasets? This will allow for a broader understanding of site-specific limitations.

*Reply* : Numerous studies have been aiming to define a soil roughness optimization based on various auxiliary datasets. For both SMAP and SMOS, $H_{r,g}$ depends on the International Geosphere-Biosphere Programme (IGBP) land cover map (Kerr et al., 2020; Chaubell et al., 2020). In the present study, we observed that decrease in $H_{r,g}$ leads to decreasing biases. A summary table of the biases was added in Appendix D with the smallest bias per site/line in bold (Table 1). However, at first glance, no clear relation appears between this and the site land cover (Table 1) or soil type (Table A1) (see R1-C3). Moreover, this work aimed to assess the faisability of $Tg$ retrievals from SMOS observations, and with limited additional data. That is why we focused on defining only one $H_{r,g}$ value to suit all sites.

**R3-C4**

Could a sensitivity analysis be added to assess how variations in permittivity affect retrieval accuracy?

*Reply* : In this manuscript, we aim to prove the faisability of retrieving $T_g$ under the snowpack from SMOS observations. As a sensitivity analysis would be a key element to improve the model and assess the uncertainties, we are currently conducting another study on this topic.

The Figure 7 below shows the sensitivity of derived ground temperature as a function of ground permittivity (real and imaginary parts as x/y axis). Each cell shows the value of $\Delta T_g = T_{g,inv} - T_{g,obs}$ in °C. The derived temperature highly depends on the permittivity $\varepsilon_g$, as $\mathrm{Re}(\varepsilon_g)$ varying from 1 to 10 leads to $\Delta T_g$ from 8.3°C to -12.6° and $\mathrm{Im}(\varepsilon_g)$ varying from 0 to 4.5 leads to $\Delta T_g$ from 8.3°C to -13.2°. The difference $\Delta T_g$ increases in the negative values with increasing real or imaginary part. A variation of 1 in permittivity in the real part (resp. imaginary part) leads to a delta of 3 K (resp. 0.4 K) in the derived temperature around our hypothesis of a ground permittivity of 5 + 0.5 i. A wise choice of the soil permittivity value appears to be crucial to a successful retrieval. We provided some elements of discussion in R2-C3.

[Figure]

Figure 7: Sensitivity analysis of the model for ground permittivity $\varepsilon_g$. The difference between "observed" $T_{g,obs}$ and inverted $T_{g,inv}$ are shown depending of the "observed" ground permittivity $\varepsilon_{g,obs}$. The ground permittivity used in the model is $\varepsilon_{g,mod} = 5 + 0.5$ i.

**R3-C5**

A more detailed uncertainty analysis, including the impact of RFI, atmospheric contributions, and snow property variability, would provide more information on the model's capabilities.

*Reply* : Indeed, RFI highly impact the SMOS observations. However, it is difficult to relate the RFI intensity, which varies a lot in space and time, to retrieval uncertainties. A strategy detection was developed by Richaume et al. (2014) to discard TB that may be impacted by RFI. Consequently, we use a RFI ratio of 0.1 as defined in Kerr et al. (2020). Previous studies (Pellarin et al., 2003; Kerr et al., 2020) have shown that temporal and spatial variations in atmospheric contributions are limited.

Figure 8 shows a sensitivity test of the impact of an error in snow density. The "observed" $T_{\mathrm{B}}$ in both polarizations (H in full line and V in dashed line) are obtained using various $\rho_{\mathrm{s,obs}}$ in our emission model (left graph). From these BT, a $T_{\mathrm{g,inv}}$ is inverted, considering our model with a snow density equal to $\rho_{\mathrm{s,inv}} = 300 \ \mathrm{kg \ m^{-3}}$. The right graph shows the difference $\Delta T_{\mathrm{g}}$ between inverted $T_{\mathrm{g,inv}}$ and "observed" $T_{\mathrm{g,obs}}$ depending of the "observed" snow density $\rho_{\mathrm{s,obs}}$. If $\rho_{\mathrm{s,obs}} < \rho_{\mathrm{s,mod}}$, $\Delta T_{\mathrm{g}} > 0$, up to 2 K. Otherwise, if the snow density is underestimated, $\Delta T_{\mathrm{g}}$ is slightly negative, down to -2 K. Derksen et al. (2014) showed that snow density across the Arctic area was between 200 and 400 $\mathrm{kg \ m^{-3}}$. The corresponding absolute $\Delta T_{\mathrm{g}}$ is lower than 0.5 K. We aim to detail similar sensitivity analysis in an ongoing study (see R2-C2).

[Figure]

Figure 8: Sensitivity analysis of the model for snow density $\rho_{\mathrm{s}}$. Left graph shows the "observed" $T_{\mathrm{B}}$ in both polarizations (H in full line and V in dashed line), using various $\rho_{\mathrm{s,obs}}$. Right graph shows the difference between "observed" $T_{\mathrm{g,obs}}$ and inverted $T_{\mathrm{g,inv}}$ depending of the "observed" snow density $\rho_{\mathrm{s,obs}}$. The snow density of the model is $\rho_{\mathrm{s,mod}} = 300 \ \mathrm{kg \ m^{-3}}$.

**R3-C6**

A table summarizing performance across all sites such as median bias, R, etc. would be helpful.

*Reply* : A summary table per metric (R, bias and ubRMSD) was added in the Appendices and introduced in the results section :

*New* (207 - 209): "The metrics (bias, R and ubRMSD) for all sites and obtained with both $\mathrm{MEM_G}$ and $\mathrm{MEM_{G+WI}}$ are summarized in Appendix D. This results section first focuses on the $H_{\mathrm{r,gs}}$ and $H_{\mathrm{r,wi}}$ optimization based on the biases (Section 4.1.) and then evaluates the $T_{\mathrm{g}}$ retrievals (Section 4.2.)."

Table 1: Biases in °C for all sites (lines) and all $H_r$ (columns). The last column gathers scores from ERA5. The sub-table on top corresponds to the model $\text{MEM}_G$ and the one bellow to the model $\text{MEM}_G$. The smallest bias obtained with $\text{MEM}_G$ or $\text{MEM}_{G+WI}$ per site (i.e. line) is in bold.

| | Bias in °C | | | | | | | | | | | |
| --- | --- | --- | --- | --- | --- | --- | --- | --- | --- | --- | --- | --- |
| | | | | | | $\text{MEM}_G$ | | | | | | ERA5 |
| $H_{r,gs}$ | 0 | 0.1 | 0.2 | 0.3 | 0.4 | 0.5 | 0.6 | 0.7 | 0.8 | 0.9 | 1 | |
| Awuna2 | 18.0 | 15.6 | 13.5 | 11.7 | 10.0 | 8.5 | 7.2 | 5.9 | 4.9 | 3.9 | **3.0** | 1.8 |
| Camden Bay | 11.5 | 9.2 | 7.2 | 5.4 | 3.7 | 2.3 | 2.0 | **-0.2** | -1.3 | -2.2 | -3.0 | 0.8 |
| East Teshekpuk | **-9.8** | -11.9 | -13.8 | -15.5 | -17.0 | -18.3 | -19.5 | -20.6 | -21.6 | -22.5 | -23.3 | 3.5 |
| Fish Creek | 6.1 | 3.9 | 1.8 | **0.1** | -1.5 | -3.0 | -4.3 | -5.4 | -6.5 | -7.4 | -8.3 | 0.6 |
| Ikpikpuk | **-6.4** | -8.6 | -10.5 | -12.2 | -13.7 | -15.1 | -16.3 | -17.4 | -18.4 | -19.3 | -20.1 | 2.3 |
| Inigok | 3.1 | **0.9** | -1.1 | -2.8 | -4.4 | -5.8 | -7.1 | -8.3 | -9.3 | -10.2 | -11.1 | 3.0 |
| Koluktak | 3.5 | 1.2 | **-0.7** | -2.5 | -4.1 | -5.5 | -6.8 | -8.0 | -9.0 | -9.9 | -10.8 | 0.8 |
| Lake 145 | **-2.1** | -4.3 | -6.2 | -8.0 | -9.5 | -10.9 | -12.2 | -13.3 | -14.3 | -15.2 | -16.0 | 3.2 |
| Marsh Creek | 9.1 | 6.9 | 4.8 | 3.0 | 1.4 | **-0.1** | -1.4 | -2.5 | -3.6 | -4.5 | -5.4 | -0.8 |
| Niguanak | 12.6 | 10.3 | 8.2 | 6.4 | 4.8 | 3.3 | 2.0 | 0.8 | **-0.3** | -1.2 | -2.1 | -1.1 |
| Piksiksak | 13.6 | 11.3 | 9.3 | 7.5 | 5.8 | 4.3 | 3.0 | 1.8 | **0.8** | -0.2 | -1.0 | 3.2 |
| South Meade | 3.3 | 1.1 | **-0.9** | -2.6 | -4.2 | -5.6 | -6.9 | -8.0 | -9.0 | -9.9 | -10.8 | 3.6 |
| Tunalik | 12.6 | 10.3 | 8.3 | 6.5 | 4.8 | 3.4 | 2.1 | 0.9 | **-0.2** | -1.1 | -2.0 | 4.8 |
| Umiat | 16.6 | 14.3 | 12.2 | 10.3 | 8.6 | 7.2 | 5.8 | 4.6 | 3.5 | 2.6 | **1.7** | 2.2 |
| Atqasuk | **-0.7** | -2.9 | -4.9 | -6.6 | -8.2 | -9.6 | -10.9 | -12.0 | -13.0 | -13.9 | -14.8 | 1.1 |
| Barrow | **-1.9** | -4.1 | -6.0 | -7.8 | -9.3 | -10.7 | -12.0 | -13.1 | -14.1 | -15.0 | -15.8 | -2.4 |
| Ivotuk | 11.4 | 9.0 | 7.0 | 5.1 | 3.5 | 2.0 | 0.6 | **-0.5** | -1.6 | -2.6 | -3.4 | -0.1 |
| Atigun Pass | 10.3 | 8.0 | 6.0 | 4.1 | 2.5 | 1.0 | **-0.3** | -1.5 | -2.5 | -3.5 | -4.4 | -1.7 |
| Ikalukrok Creek | 11.8 | 9.5 | 7.4 | 5.6 | 4.0 | 2.5 | 1.2 | **0.0** | -1.1 | -2.1 | -2.9 | -0.2 |
| Imnaviat Creek | 13.9 | 11.6 | 9.5 | 7.7 | 6.0 | 4.5 | 3.1 | 1.9 | 0.8 | -0.1 | -1.0 | -4.3 |
| Kelly Station | 10.1 | 7.7 | 5.7 | 3.8 | 2.2 | 0.7 | **-0.6** | -1.8 | -2.9 | -3.8 | -4.8 | -13.1 |

| | Bias in °C | | | | | | | | | | | |
| --- | --- | --- | --- | --- | --- | --- | --- | --- | --- | --- | --- | --- |
| | | | | | | $\text{MEM}_{G+WI}$ | | | | | | ERA5 |
| $H_{r,wi}$ | 0 | 0.1 | 0.2 | 0.3 | 0.4 | 0.5 | 0.6 | 0.7 | 0.8 | 0.9 | 1 | |
| Awuna2 | 4.9 | 4.9 | 4.9 | 4.9 | 4.9 | 4.9 | 4.9 | 4.9 | 4.9 | 4.9 | 4.9 | 1.8 |
| Camden Bay | 79.7 | 70.5 | 62.1 | 54.6 | 47.8 | 41.7 | 36.2 | 31.3 | 26.8 | 22.7 | **19.1** | 0.8 |
| East Teshekpuk | 37.0 | 28.8 | 21.4 | 14.8 | 8.8 | 3.4 | **-1.5** | -5.9 | -9.8 | -13.4 | -16.7 | 3.5 |
| Fish Creek | 137.4 | 120.4 | 105.1 | 91.4 | 79.0 | 67.7 | 57.6 | 48.5 | 40.3 | 32.9 | **26.2** | 0.6 |
| Ikpikpuk | 22.7 | 17.1 | 12.1 | 7.6 | 3.6 | -0.1 | -3.4 | -6.4 | -9.1 | -11.5 | -13.7 | 2.3 |
| Inigok | 19.9 | 16.4 | 13.2 | 10.3 | 7.8 | 5.4 | 3.4 | 1.5 | **-0.2** | -1.8 | -3.2 | 3.0 |
| Koluktak | 15.6 | 12.7 | 10.0 | 7.7 | 5.5 | 3.6 | 1.8 | **0.2** | -1.2 | -2.5 | -3.7 | 0.8 |
| Lake 145 | 44.4 | 36.8 | 30.0 | 23.9 | 18.4 | 13.4 | 9.0 | 4.9 | **1.3** | -2.0 | -5.0 | 3.2 |
| Marsh Creek | 77.4 | 68.2 | 59.8 | 52.3 | 45.5 | 39.4 | 33.9 | 29.0 | 24.5 | 20.4 | **16.8** | -0.8 |
| Niguanak | 29.6 | 26.3 | 23.3 | 20.6 | 18.1 | 16.0 | 14.0 | 12.2 | 10.6 | 9.1 | **7.8** | -1.1 |
| Piksiksak | 5.2 | 4.7 | 4.3 | 3.9 | 3.5 | 3.2 | 2.9 | 2.6 | 2.4 | 2.2 | **2.0** | 3.2 |
| South Meade | 25.6 | 21.2 | 17.3 | 13.8 | 10.6 | 7.7 | 5.1 | 2.8 | **0.7** | -1.2 | -3.0 | 3.6 |
| Tunalik | 2.0 | 1.7 | 1.5 | 1.3 | 1.1 | 1.0 | 0.8 | 0.7 | 0.6 | 0.5 | **0.4** | 4.8 |
| Umiat | 5.8 | 5.6 | 5.3 | 5.2 | 5.0 | 4.8 | 4.7 | 4.5 | 4.4 | 4.3 | **4.2** | 2.2 |
| Atqasuk | 16.5 | 12.8 | 9.5 | 6.5 | 3.7 | 1.3 | **-0.9** | -2.9 | -4.7 | -6.3 | -7.8 | 1.1 |
| Barrow | 29.3 | 23.7 | 18.7 | 14.2 | 10.2 | 6.5 | 3.2 | **0.2** | -2.5 | -4.9 | -7.1 | -2.4 |
| Ivotuk | -1.6 | -1.6 | -1.6 | -1.6 | -1.6 | -1.6 | -1.6 | -1.6 | -1.6 | -1.6 | -1.6 | -0.1 |
| Atigun Pass | **-1.5** | -1.6 | -1.7 | -1.8 | -1.9 | -1.9 | -2.0 | -2.1 | -2.1 | -2.2 | -2.2 | -1.7 |
| Ikalukrok Creek | -1.1 | -1.1 | -1.1 | -1.1 | -1.1 | -1.1 | -1.1 | -1.1 | -1.1 | -1.1 | -1.1 | -0.2 |
| Imnaviat Creek | 2.0 | 1.9 | 1.7 | 1.7 | 1.6 | 1.5 | 1.4 | 1.4 | 1.3 | **1.2** | **1.2** | -4.3 |
| Kelly Station | 0.5 | **0.1** | -0.2 | -0.5 | -0.7 | -1.0 | -1.2 | -1.4 | -1.6 | -1.8 | -1.9 | -13.1 |

Table 2: R for all sites(lines) and all $H_r$ (columns). The last column gathers scores from ERA5. The sub-table on top corresponds to the model MEM$_G$ and the one bellow to the model MEM$_G$.

| | R | | | | | | | | | | | ERA5 |
|---|---|---|---|---|---|---|---|---|---|---|---|---|
| | MEM$_G$ | | | | | | | | | | | |
| $H_{r,gs}$ | 0 | 0.1 | 0.2 | 0.3 | 0.4 | 0.5 | 0.6 | 0.7 | 0.8 | 0.9 | 1 | |
| Awuna2 | 0.64 | 0.63 | 0.63 | 0.63 | 0.63 | 0.63 | 0.63 | 0.63 | 0.63 | 0.63 | 0.63 | 0.55 |
| Camden Bay | 0.30 | 0.30 | 0.29 | 0.29 | 0.30 | 0.30 | 0.29 | 0.29 | 0.29 | 0.29 | 0.29 | 0.78 |
| East Teshekpuk | 0.13 | 0.13 | 0.13 | 0.13 | 0.13 | 0.13 | 0.12 | 0.13 | 0.12 | 0.13 | 0.13 | 0.58 |
| Fish Creek | 0.24 | 0.24 | 0.23 | 0.23 | 0.24 | 0.23 | 0.23 | 0.24 | 0.24 | 0.23 | 0.23 | 0.74 |
| Ikpikpuk | 0.35 | 0.35 | 0.35 | 0.35 | 0.35 | 0.35 | 0.35 | 0.35 | 0.35 | 0.35 | 0.35 | 0.70 |
| Inigok | 0.42 | 0.42 | 0.42 | 0.42 | 0.42 | 0.42 | 0.42 | 0.42 | 0.42 | 0.42 | 0.42 | 0.54 |
| Koluktak | 0.45 | 0.45 | 0.45 | 0.45 | 0.45 | 0.45 | 0.45 | 0.45 | 0.44 | 0.45 | 0.44 | 0.63 |
| Lake 145 | 0.16 | 0.15 | 0.15 | 0.15 | 0.15 | 0.15 | 0.15 | 0.15 | 0.15 | 0.15 | 0.15 | 0.61 |
| Marsh Creek | 0.23 | 0.23 | 0.23 | 0.23 | 0.23 | 0.22 | 0.22 | 0.22 | 0.22 | 0.23 | 0.22 | 0.63 |
| Niguanak | 0.42 | 0.42 | 0.42 | 0.42 | 0.41 | 0.41 | 0.41 | 0.41 | 0.41 | 0.41 | 0.41 | 0.79 |
| Piksiksak | 0.74 | 0.74 | 0.74 | 0.74 | 0.73 | 0.74 | 0.74 | 0.74 | 0.74 | 0.74 | 0.74 | 0.51 |
| South Meade | 0.22 | 0.21 | 0.21 | 0.21 | 0.21 | 0.21 | 0.21 | 0.21 | 0.21 | 0.21 | 0.21 | 0.53 |
| Tunalik | 0.75 | 0.75 | 0.75 | 0.75 | 0.75 | 0.75 | 0.75 | 0.75 | 0.75 | 0.75 | 0.75 | 0.55 |
| Umiat | 0.60 | 0.60 | 0.60 | 0.60 | 0.60 | 0.60 | 0.60 | 0.59 | 0.60 | 0.60 | 0.59 | 0.62 |
| Atqasuk | -0.24 | -0.24 | -0.24 | -0.23 | -0.23 | -0.23 | -0.23 | -0.23 | -0.24 | -0.24 | -0.24 | 0.40 |
| Barrow | 0.05 | 0.05 | 0.05 | 0.05 | 0.06 | 0.06 | 0.06 | 0.06 | 0.06 | 0.06 | 0.06 | 0.36 |
| Ivotuk | 0.50 | 0.49 | 0.50 | 0.49 | 0.50 | 0.49 | 0.49 | 0.49 | 0.49 | 0.49 | 0.50 | -0.05 |
| Atigun Pass | 0.68 | 0.68 | 0.68 | 0.68 | 0.68 | 0.68 | 0.68 | 0.68 | 0.68 | 0.68 | 0.68 | 0.72 |
| Ikalukrok Creek | 0.51 | 0.51 | 0.51 | 0.51 | 0.51 | 0.51 | 0.51 | 0.51 | 0.51 | 0.51 | 0.51 | -0.09 |
| Imnaviat Creek | 0.55 | 0.55 | 0.54 | 0.55 | 0.54 | 0.54 | 0.54 | 0.54 | 0.54 | 0.54 | 0.54 | 0.50 |
| Kelly Station | 0.41 | 0.40 | 0.40 | 0.39 | 0.41 | 0.41 | 0.41 | 0.40 | 0.38 | 0.41 | 0.39 | 0.33 |

| | R | | | | | | | | | | | ERA5 |
|---|---|---|---|---|---|---|---|---|---|---|---|---|
| | MEM$_{G+WI}$ | | | | | | | | | | | |
| $H_{r,wi}$ | 0 | 0.1 | 0.2 | 0.3 | 0.4 | 0.5 | 0.6 | 0.7 | 0.8 | 0.9 | 1 | |
| Awuna2 | 0.63 | 0.63 | 0.63 | 0.63 | 0.63 | 0.63 | 0.63 | 0.63 | 0.63 | 0.63 | 0.63 | 0.55 |
| Camden Bay | 0.29 | 0.29 | 0.29 | 0.29 | 0.29 | 0.29 | 0.29 | 0.29 | 0.29 | 0.29 | 0.29 | 0.78 |
| East Teshekpuk | 0.13 | 0.13 | 0.13 | 0.13 | 0.13 | 0.13 | 0.13 | 0.13 | 0.13 | 0.13 | 0.13 | 0.58 |
| Fish Creek | 0.24 | 0.24 | 0.24 | 0.24 | 0.24 | 0.24 | 0.24 | 0.24 | 0.24 | 0.24 | 0.24 | 0.74 |
| Ikpikpuk | 0.35 | 0.35 | 0.35 | 0.35 | 0.35 | 0.36 | 0.35 | 0.35 | 0.35 | 0.35 | 0.35 | 0.70 |
| Inigok | 0.41 | 0.41 | 0.41 | 0.41 | 0.42 | 0.42 | 0.42 | 0.42 | 0.42 | 0.42 | 0.41 | 0.54 |
| Koluktak | 0.44 | 0.45 | 0.44 | 0.44 | 0.45 | 0.44 | 0.45 | 0.45 | 0.44 | 0.45 | 0.44 | 0.63 |
| Lake 145 | 0.16 | 0.15 | 0.16 | 0.15 | 0.15 | 0.15 | 0.15 | 0.15 | 0.15 | 0.15 | 0.15 | 0.61 |
| Marsh Creek | 0.23 | 0.23 | 0.23 | 0.24 | 0.23 | 0.23 | 0.22 | 0.22 | 0.22 | 0.23 | 0.22 | 0.63 |
| Niguanak | 0.41 | 0.41 | 0.41 | 0.41 | 0.41 | 0.41 | 0.41 | 0.41 | 0.41 | 0.41 | 0.41 | 0.79 |
| Piksiksak | 0.74 | 0.74 | 0.74 | 0.74 | 0.74 | 0.74 | 0.74 | 0.74 | 0.74 | 0.74 | 0.74 | 0.51 |
| South Meade | 0.21 | 0.21 | 0.21 | 0.21 | 0.21 | 0.21 | 0.21 | 0.21 | 0.21 | 0.21 | 0.21 | 0.53 |
| Tunalik | 0.75 | 0.75 | 0.75 | 0.75 | 0.75 | 0.75 | 0.75 | 0.75 | 0.75 | 0.75 | 0.75 | 0.55 |
| Umiat | 0.60 | 0.60 | 0.60 | 0.60 | 0.60 | 0.59 | 0.60 | 0.60 | 0.60 | 0.60 | 0.60 | 0.62 |
| Atqasuk | -0.24 | -0.24 | -0.24 | -0.24 | -0.24 | -0.24 | -0.24 | -0.24 | -0.24 | -0.24 | -0.24 | 0.40 |
| Barrow | 0.06 | 0.06 | 0.06 | 0.06 | 0.06 | 0.06 | 0.06 | 0.06 | 0.06 | 0.06 | 0.06 | 0.36 |
| Ivotuk | 0.49 | 0.49 | 0.49 | 0.49 | 0.49 | 0.49 | 0.49 | 0.49 | 0.49 | 0.49 | 0.49 | -0.05 |
| Atigun Pass | 0.68 | 0.68 | 0.68 | 0.68 | 0.68 | 0.68 | 0.68 | 0.68 | 0.68 | 0.68 | 0.68 | 0.72 |
| Ikalukrok Creek | 0.51 | 0.51 | 0.51 | 0.51 | 0.51 | 0.51 | 0.51 | 0.51 | 0.51 | 0.51 | 0.51 | -0.09 |
| Imnaviat Creek | 0.54 | 0.54 | 0.54 | 0.54 | 0.54 | 0.54 | 0.54 | 0.54 | 0.54 | 0.54 | 0.54 | 0.50 |
| Kelly Station | 0.42 | 0.41 | 0.41 | 0.41 | 0.43 | 0.40 | 0.40 | 0.41 | 0.40 | 0.40 | 0.40 | 0.33 |

Table 3: ubRMSD in °C for all sites (lines) and all $H_\mathrm{r}$ (columns). The last column gathers scores from ERA5. The sub-table on top corresponds to the model $\mathrm{MEM_G}$ and the one bellow to the model $\mathrm{MEM_G}$.

| | ubRMSD in °C | | | | | | | | | | | ERA5 |
| | $\mathrm{MEM_G}$ | | | | | | | | | | | |
| $H_\mathrm{r,gs}$ | 0 | 0.1 | 0.2 | 0.3 | 0.4 | 0.5 | 0.6 | 0.7 | 0.8 | 0.9 | 1 | |
|---|---|---|---|---|---|---|---|---|---|---|---|---|
| Awuna2 | 2.3 | 2.3 | 2.3 | 2.3 | 2.3 | 2.3 | 2.3 | 2.3 | 2.3 | 2.3 | 2.3 | 3.2 |
| Camden Bay | 4.6 | 4.6 | 4.6 | 4.6 | 4.5 | 4.5 | 4.5 | 4.5 | 4.5 | 4.5 | 4.5 | 2.7 |
| East Teshekpuk | 4.4 | 4.3 | 4.3 | 4.3 | 4.3 | 4.3 | 4.3 | 4.3 | 4.2 | 4.2 | 4.2 | 3.5 |
| Fish Creek | 3.8 | 3.8 | 3.8 | 3.8 | 3.8 | 3.8 | 3.7 | 3.7 | 3.7 | 3.7 | 3.7 | 2.3 |
| Ikpikpuk | 3.8 | 3.8 | 3.8 | 3.8 | 3.8 | 3.8 | 3.8 | 3.8 | 3.8 | 3.8 | 3.8 | 3.2 |
| Inigok | 2.9 | 2.9 | 2.9 | 2.9 | 2.9 | 2.9 | 2.9 | 2.9 | 2.9 | 2.9 | 2.9 | 3.4 |
| Koluktak | 2.9 | 2.9 | 2.9 | 2.9 | 2.9 | 2.9 | 2.9 | 2.9 | 2.9 | 2.9 | 2.9 | 3.1 |
| Lake 145 | 4.3 | 4.3 | 4.3 | 4.2 | 4.2 | 4.2 | 4.2 | 4.2 | 4.2 | 4.2 | 4.2 | 3.3 |
| Marsh Creek | 4.4 | 4.4 | 4.4 | 4.4 | 4.4 | 4.4 | 4.4 | 4.4 | 4.3 | 4.3 | 4.3 | 2.9 |
| Niguanak | 3.8 | 3.8 | 3.8 | 3.7 | 3.7 | 3.7 | 3.7 | 3.7 | 3.7 | 3.7 | 3.7 | 2.6 |
| Piksiksak | 2.5 | 2.5 | 2.5 | 2.5 | 2.5 | 2.5 | 2.5 | 2.5 | 2.5 | 2.5 | 2.5 | 4.1 |
| South Meade | 5.4 | 5.4 | 5.4 | 5.4 | 5.4 | 5.3 | 5.3 | 5.3 | 5.3 | 5.3 | 5.3 | 4.0 |
| Tunalik | 3.1 | 3.1 | 3.1 | 3.1 | 3.1 | 3.1 | 3.1 | 3.1 | 3.1 | 3.1 | 3.1 | 4.4 |
| Umiat | 2.1 | 2.1 | 2.1 | 2.1 | 2.1 | 2.1 | 2.1 | 2.1 | 2.1 | 2.1 | 2.1 | 3.2 |
| Atqasuk | 5.8 | 5.8 | 5.7 | 5.7 | 5.7 | 5.7 | 5.7 | 5.6 | 5.6 | 5.6 | 5.6 | 4.1 |
| Barrow | 8.3 | 8.2 | 8.2 | 8.2 | 8.1 | 8.1 | 8.1 | 8.0 | 8.0 | 8.0 | 8.0 | 3.5 |
| Ivotuk | 2.1 | 2.1 | 2.0 | 2.0 | 2.0 | 2.0 | 2.0 | 2.0 | 2.0 | 2.0 | 2.0 | 3.9 |
| Atigun Pass | 1.5 | 1.5 | 1.5 | 1.5 | 1.5 | 1.5 | 1.5 | 1.4 | 1.5 | 1.4 | 1.4 | 2.0 |
| Ikalukrok Creek | 3.0 | 3.0 | 3.0 | 3.0 | 3.0 | 3.0 | 3.0 | 3.0 | 3.0 | 3.0 | 3.0 | 7.2 |
| Imnaviat Creek | 1.4 | 1.4 | 1.3 | 1.3 | 1.3 | 1.3 | 1.3 | 1.3 | 1.3 | 1.3 | 1.3 | 2.2 |
| Kelly Station | 1.5 | 1.5 | 1.5 | 1.5 | 1.5 | 1.5 | 1.5 | 1.5 | 1.5 | 1.4 | 1.4 | 6.0 |

| | ubRMSD in °C | | | | | | | | | | | ERA5 |
| | $\mathrm{MEM_{G+WI}}$ | | | | | | | | | | | |
| $H_\mathrm{r,wi}$ | 0 | 0.1 | 0.2 | 0.3 | 0.4 | 0.5 | 0.6 | 0.7 | 0.8 | 0.9 | 1 | |
|---|---|---|---|---|---|---|---|---|---|---|---|---|
| Awuna2 | 2.3 | 2.3 | 2.3 | 2.3 | 2.3 | 2.3 | 2.3 | 2.3 | 2.3 | 2.3 | 2.3 | 3.2 |
| Camden Bay | 6.8 | 6.8 | 6.8 | 6.7 | 6.8 | 6.8 | 6.8 | 6.8 | 6.7 | 6.7 | 6.7 | 2.7 |
| East Teshekpuk | 6.1 | 6.1 | 6.1 | 6.2 | 6.1 | 6.1 | 6.1 | 6.1 | 6.1 | 6.1 | 6.1 | 3.5 |
| Fish Creek | 7.8 | 7.7 | 7.7 | 7.7 | 7.7 | 7.7 | 7.7 | 7.7 | 7.7 | 7.7 | 7.7 | 2.3 |
| Ikpikpuk | 4.4 | 4.4 | 4.4 | 4.4 | 4.4 | 4.4 | 4.4 | 4.4 | 4.4 | 4.4 | 4.4 | 3.2 |
| Inigok | 3.3 | 3.3 | 3.3 | 3.3 | 3.3 | 3.3 | 3.3 | 3.3 | 3.3 | 3.3 | 3.3 | 3.4 |
| Koluktak | 3.2 | 3.2 | 3.2 | 3.2 | 3.2 | 3.2 | 3.2 | 3.2 | 3.2 | 3.2 | 3.2 | 3.1 |
| Lake 145 | 6.0 | 6.0 | 6.0 | 6.0 | 6.0 | 6.0 | 6.0 | 6.0 | 6.0 | 6.0 | 6.0 | 3.3 |
| Marsh Creek | 7.1 | 7.1 | 7.1 | 7.0 | 7.1 | 7.1 | 7.1 | 7.1 | 7.1 | 7.1 | 7.1 | 2.9 |
| Niguanak | 4.4 | 4.4 | 4.4 | 4.4 | 4.4 | 4.4 | 4.4 | 4.4 | 4.4 | 4.4 | 4.4 | 2.6 |
| Piksiksak | 2.5 | 2.5 | 2.5 | 2.5 | 2.5 | 2.5 | 2.5 | 2.5 | 2.5 | 2.5 | 2.5 | 4.1 |
| South Meade | 6.5 | 6.5 | 6.6 | 6.5 | 6.6 | 6.6 | 6.6 | 6.6 | 6.6 | 6.6 | 6.6 | 4.0 |
| Tunalik | 3.1 | 3.1 | 3.1 | 3.1 | 3.1 | 3.1 | 3.1 | 3.1 | 3.1 | 3.1 | 3.1 | 4.4 |
| Umiat | 2.1 | 2.1 | 2.1 | 2.1 | 2.1 | 2.1 | 2.1 | 2.1 | 2.1 | 2.1 | 2.1 | 3.2 |
| Atqasuk | 6.7 | 6.7 | 6.7 | 6.7 | 6.7 | 6.7 | 6.7 | 6.7 | 6.7 | 6.7 | 6.7 | 4.1 |
| Barrow | 11.5 | 11.5 | 11.5 | 11.5 | 11.5 | 11.5 | 11.5 | 11.5 | 11.5 | 11.5 | 11.5 | 3.5 |
| Ivotuk | 2.0 | 2.0 | 2.0 | 2.0 | 2.0 | 2.0 | 2.0 | 2.0 | 2.0 | 2.0 | 2.0 | 3.9 |
| Atigun Pass | 1.5 | 1.5 | 1.5 | 1.5 | 1.5 | 1.5 | 1.5 | 1.5 | 1.5 | 1.5 | 1.5 | 2.0 |
| Ikalukrok Creek | 3.0 | 3.0 | 3.0 | 3.0 | 3.0 | 3.0 | 3.0 | 3.0 | 3.0 | 3.0 | 3.0 | 7.2 |
| Imnaviat Creek | 1.3 | 1.3 | 1.3 | 1.4 | 1.4 | 1.3 | 1.3 | 1.4 | 1.3 | 1.3 | 1.3 | 2.2 |
| Kelly Station | 1.5 | 1.5 | 1.5 | 1.5 | 1.5 | 1.5 | 1.5 | 1.5 | 1.5 | 1.5 | 1.5 | 6.0 |

**R3-C7**

The assumptions underlying the models such as constant ground permittivity, snow transparency etc. should be clearly stated and justified in the Methods section.

*Reply* : Details about constant ground permittivity and other hypothesis can be find in R2-C2 and R2-C3. We also precised the assumption behind the term "snow transparency" as:

*New (158 -160)* : "According to Schwank et al. (2015) and Rautiainen et al. (2016), dry snow *is* considered transparent at L-band, i.e. its internal transmissivity and reflectivity are $t_\mathrm{s} = 1$ *(no absorption)* and $r_\mathrm{s} = 0$ *(no volume scattering)*".

Some figures like 5 – 7 are generally a bit overly complicated, as is the colour scheme. Try to simplify.

*Reply* : Indeed, Figures 5/6/7 displayed a lot of information. As it was probably too complicated, we simplified these figures as recommended (Figures 2, 3, 4, see R2-C4). However, we consider the confidence intervals to be useful for data analysis and interpretation, so we kept it in appendix figures (Appendix B: Results: Figures with confidence intervals).

Figure 8 is a really interesting figure for this paper. It could be improved by adding a legend or annotations to clarify the different lines. I understand shading the text to represent it, but this might not to be intuitive to readers.

*Reply* : We modified the Figure 8 to make it clearer. The whole legend is now bellow the figure, and the caption was modified to indicate the axis meaning (see R2-C5).

**References**

Adams, W. and Lasenby, D.: The Roles of Snow, Lake Ice and Lake Water in the Distribution of Major Ions in the Ice Cover of a Lake, Annals of Glaciology, 7, 202–207, doi:10.3189/S0260305500006170, 1985.

Biancamaria, S., Lettenmaier, D. P., and Pavelsky, T. M.: The SWOT Mission and Its Capabilities for Land Hydrology, in: Remote Sensing and Water Resources, edited by Cazenave, A., Champollion, N., Benveniste, J., and Chen, J., vol. 55, pp. 117–147, Springer International Publishing, Cham, ISBN 978-3-319-32448-7 978-3-319-32449-4, doi:10.1007/978-3-319-32449-4_6, 2016.

Chaubell, M. J., Yueh, S. H., Dunbar, R. S., Colliander, A., Chen, F., Chan, S. K., Entekhabi, D., Bindlish, R., O'Neill, P. E., Asanuma, J., Berg, A. A., Bosch, D. D., Caldwell, T., Cosh, M. H., Holifield Collins, C., Martinez-Fernandez, J., Seyfried, M., Starks, P. J., Su, Z., Thibeault, M., and Walker, J.: Improved SMAP Dual-Channel Algorithm for the Retrieval of Soil Moisture, IEEE Transactions on Geoscience and Remote Sensing, 58, 3894–3905, doi:10.1109/TGRS.2019.2959239, 2020.

Derksen, C., Lemmetyinen, J., Toose, P., Silis, A., Pulliainen, J., and Sturm, M.: Physical Properties of Arctic versus Subarctic Snow: Implications for High Latitude Passive Microwave Snow Water Equivalent Retrievals, Journal of Geophysical Research: Atmospheres, 119, 7254–7270, doi:10.1002/2013JD021264, 2014.

Dolant, C., Langlois, A., Brucker, L., Royer, A., Roy, A., and Montpetit, B.: Meteorological Inventory of Rain-on-Snow Events in the Canadian Arctic Archipelago and Satellite Detection Assessment Using Passive Microwave Data, Physical Geography, 39, 428–444, doi:10.1080/02723646.2017.1400339, 2018.

Domine, F., Fourteau, K., Picard, G., Lackner, G., Sarrazin, D., and Poirier, M.: Permafrost Cooled in Winter by Thermal Bridging through Snow-Covered Shrub Branches, Nature Geoscience, 15, 554–560, doi:10.1038/s41561-022-00979-2, 2022.

Donlon, C., Galeazzi, C., Midthassel, R., Sallusti, M., Triggianese, M., Fiorelli, B., De Paris, G., Kornienko, A., and Khlystova, I.: The Copernicus Imaging Microwave Radiometer (CIMR): Mission Overview and Status, in: IGARSS 2023 - 2023 IEEE International Geoscience and Remote Sensing Symposium, pp. 989–992, IEEE, Pasadena, CA, USA, ISBN 9798350320107, doi:10.1109/IGARSS52108.2023.10281934, 2023.

Duguay, C. R. and Lafleur, P. M.: Determining Depth and Ice Thickness of Shallow Sub-Arctic Lakes Using Space-Borne Optical and SAR Data, International Journal of Remote Sensing, 24, 475–489, doi:10.1080/01431160304992, 2003.

Gélinas, A., Filali, B., Langlois, A., Kelly, R., Mavrovic, A., Demontoux, F., and Roy, A.: New Wideband Large Aperture Open-Ended Coaxial Microwave Probe for Soil Dielectric Characterization, IEEE Transactions on Geoscience and Remote Sensing, pp. 1–1, doi:10.1109/TGRS.2025.3539532, 2025.

Gibon, F., Mialon, A., Richaume, P., Kerr, Y. H., Rodriguez-Fernandez, N. J., Sabia, R., and Boresch, A.: WORK PACKAGE 5 TECHNICAL NOTE 1 SO-TN-CB-GS-0104 REQ-16: R&D CASE STUDY 1 SMOS VALIDATION AND COMMITTED AREAS, 2022.

Gibon, F., Mialon, A., Richaume, P., Rodríguez-Fernández, N., Aberer, D., Boresch, A., Crapolicchio, R., Dorigo, W., Gruber, A., Himmelbauer, I., Preimesberger, W., Sabia, R., Stradiotti, P., Tercjak, M., and Kerr, Y. H.: Estimating the Uncertainties of Satellite Derived Soil Moisture at Global Scale, Science of Remote Sensing, 10, 100 147, doi:10.1016/j.srs.2024.100147, 2024.

Grenfell, T. C. and Putkonen, J.: A Method for the Detection of the Severe Rain-on-Snow Event on Banks Island, October 2003, Using Passive Microwave Remote Sensing: MICROWAVE DETECTION OF RAIN ON SNOW, Water Resources Research, 44, doi:10.1029/2007WR005929, 2008.

Grünberg, I., Wilcox, E. J., Zwieback, S., Marsh, P., and Boike, J.: Linking Tundra Vegetation, Snow, Soil Temperature, and Permafrost, Biogeosciences, 17, 4261–4279, doi:10.5194/bg-17-4261-2020, 2020.

Gutierrez, A., Castro, R., Barbosa, J., and Anterrieu, E.: Review of the Image Reconstruction Techniques Used in SMOS Data Processing, in: 2012 IEEE International Geoscience and Remote Sensing Symposium, pp. 4648–4651, IEEE, Munich, Germany, ISBN 978-1-4673-1159-5 978-1-4673-1160-1 978-1-4673-1158-8, doi:10.1109/IGARSS.2012.6350429, 2012.

Holmberg, M., Lemmetyinen, J., Schwank, M., Kontu, A., Rautiainen, K., Merkouriadi, I., and Tamminen, J.: Retrieval of Ground, Snow, and Forest Parameters from Space Borne Passive L Band Observations. A Case Study over Sodankylä, Finland, Remote Sensing of Environment, 306, 114 143, doi:10.1016/j.rse.2024.114143, 2024.

Jones, L. A., Kimball, J. S., Reichle, R. H., Madani, N., Glassy, J., Ardizzone, J. V., Colliander, A., Cleverly, J., Desai, A. R., Eamus, D., Euskirchen, E. S., Hutley, L., Macfarlane, C., and Scott, R. L.: The SMAP Level 4 Carbon Product for Monitoring Ecosystem Land–Atmosphere $CO_2$ Exchange, IEEE Transactions on Geoscience and Remote Sensing, 55, 6517–6532, doi:10.1109/TGRS.2017.2729343, 2017.

Kerr, Y., Sécherre, F., Gathier, R., and Wigneron, J. P.: Analysis of Topography Effect, Complement to ESA, ITT-3552 (CNN), Tech. rep., 2004.

Kerr, Y., Richaume, P., Waldteufel, P., Ferrazzoli, P., Wigneron, J. P., Schwank, M., and Rautiainen, K.: Algorithm Theoretical Basis Document ({ATBD}) for the SMOS Level 2 Soil Moisture Processor, Technical Report TN-ESL-SM-GS-0001-4b SM-ESL (CBSA), p. 145, 2020.

Klein, I., Uereyen, S., Sogno, P., Twele, A., Hirner, A., and Kuenzer, C.: Global WaterPack - The Development of Global Surface Water over the Past 20 Years at Daily Temporal Resolution, Scientific Data, 11, 472, doi:10.1038/s41597-024-03328-7, 2024.

Kumawat, D., Olyaei, M., Gao, L., and Ebtehaj, A.: Passive Microwave Retrieval of Soil Moisture Below Snowpack at L-Band Using SMAP Observations, IEEE Transactions on Geoscience and Remote Sensing, 60, 1–16, doi:10.1109/TGRS.2022.3216324, 2022.

Lemmetyinen, J., Kontu, A., Kärnä, J.-P., Vehviläinen, J., Takala, M., and Pulliainen, J.: Correcting for the Influence of Frozen Lakes in Satellite Microwave Radiometer Observations through Application of a Microwave Emission Model, Remote Sensing of Environment, 115, 3695–3706, doi:10.1016/j.rse.2011.09.008, 2011.

Liston, G. E., Mcfadden, J. P., Sturm, M., and Pielke, R. A.: Modelled Changes in Arctic Tundra Snow, Energy and Moisture Fluxes Due to Increased Shrubs, Global Change Biology, 8, 17–32, doi:10.1046/j.1354-1013.2001.00416.x, 2002.

Macelloni, G., Brogioni, M., Leduc-Leballeur, M., Montomoli, F., Bartsch, A., Mialon, A., Ritz, C., Soteras, J. C., Stammer, D., Picard, G., De Carolis, G., Boutin, J., Johnson, J. T., Nicholls, K. W., Jezek, K., Rautiainen, K., Kaleschke, L., Bertino, L., Tsang, L., Van Den Broeke, M., Skou, N., and Tietsche, S.: Cryorad: A Low Frequency Wideband Radiometer Mission for the Study of the Cryosphere, in: IGARSS 2018 - 2018 IEEE International Geoscience and Remote Sensing Symposium, pp. 1998–2000, IEEE, Valencia, ISBN 978-1-5386-7150-4, doi:10.1109/IGARSS.2018.8519172, 2018.

Martha Raynolds: Raster Circumpolar Arctic Vegetation Map, doi:10.17632/C4XJ5RV6KV.2, 2022.

Mätzler, C.: Passive Microwave Signatures of Landscapes in Winter, Meteorology and Atmospheric Physics, 54, 241–260, doi:10.1007/BF01030063, 1994.

Mätzler, C., ed.: Thermal Microwave Radiation: Applications for Remote Sensing, no. 52 in IET Electromagnetic Waves Series, IET, London, ISBN 978-0-86341-573-9 978-1-84919-002-2, doi:10.1049/PBEW052E, 2006.

Mialon, A., Coret, L., Kerr, Y., Secherre, F., and Wigneron, J.-P.: Flagging the Topographic Impact on the SMOS Signal, IEEE Transactions on Geoscience and Remote Sensing, 46, 689–694, doi:10.1109/TGRS.2007.914788, 2008.

Mironov, V. L., Kosolapova, L. G., Savin, I. V., and Muzalevskiy, K. V.: Temperature Dependent Dielectric Model at 1.4 GHz for a Tundra Organic-Rich Soil Thawed and Frozen, in: 2015 IEEE International Geoscience and Remote Sensing Symposium (IGARSS), pp. 2016–2019, IEEE, Milan, Italy, ISBN 978-1-4799-7929-5, doi:10.1109/IGARSS.2015.7326194, 2015.

Molero, B., Merlin, O., Malbéteau, Y., Al Bitar, A., Cabot, F., Stefan, V., Kerr, Y., Bacon, S., Cosh, M., Bindlish, R., and Jackson, T.: SMOS Disaggregated Soil Moisture Product at 1 Km Resolution: Processor Overview and First Validation Results, Remote Sensing of Environment, 180, 361–376, doi:10.1016/j.rse.2016.02.045, 2016.

Murfitt, J., Duguay, C., Picard, G., and Gunn, G.: Forward Modelling of Synthetic Aperture Radar Backscatter from Lake Ice over Canadian Subarctic Lakes, Remote Sensing of Environment, 286, 113 424, doi:10.1016/j.rse.2022.113424, 2023.

Naderpour, R., Schwank, M., and Mätzler, C.: Davos-Laret Remote Sensing Field Laboratory: 2016/2017 Winter Season L-Band Measurements Data-Processing and Analysis, Remote Sensing, 9, 1185, doi:10.3390/rs9111185, 2017a.

Naderpour, R., Schwank, M., Matzler, C., Lemmetyinen, J., and Steffen, K.: Snow Density and Ground Permittivity Retrieved From L-Band Radiometry: A Retrieval Sensitivity Analysis, IEEE Journal of Selected Topics in Applied Earth Observations and Remote Sensing, 10, 3148–3161, doi:10.1109/JSTARS.2017.2669336, 2017b.

Pekel, J.-F., Cottam, A., Gorelick, N., and Belward, A. S.: High-Resolution Mapping of Global Surface Water and Its Long-Term Changes, Nature, 540, 418–422, doi:10.1038/nature20584, 2016.

Pellarin, T., Wigneron, J.-P., Calvet, J.-C., Berger, M., Douville, H., Ferrazzoli, P., Kerr, Y., Lopez-Baeza, E., Pulliainen, J., Simmonds, L., and Waldteufel, P.: Two-Year Global Simulation of L-band Brightness Temperatures over Land, IEEE Transactions on Geoscience and Remote Sensing, 41, 2135–2139, doi:10.1109/TGRS.2003.815417, 2003.

Rautiainen, K., Lemmetyinen, J., Pulliainen, J., Vehvilainen, J., Drusch, M., Kontu, A., Kainulainen, J., and Seppanen, J.: L-Band Radiometer Observations of Soil Processes in Boreal and Subarctic Environments, IEEE Transactions on Geoscience and Remote Sensing, 50, 1483–1497, doi:10.1109/TGRS.2011.2167755, 2012.

Rautiainen, K., Parkkinen, T., Lemmetyinen, J., Schwank, M., Wiesmann, A., Ikonen, J., Derksen, C., Davydov, S., Davydova, A., Boike, J., Langer, M., Drusch, M., and Pulliainen, J.: SMOS Prototype Algorithm for Detecting Autumn Soil Freezing, Remote Sensing of Environment, 180, 346–360, doi:10.1016/j.rse.2016.01.012, 2016.

Richaume, P., Soldo, Y., Anterrieu, E., Khazaal, A., Bircher, S., Mialon, A., Bitar, A. A., Rodriguez-Fernandez, N., Cabot, F., Kerr, Y., and Mahmoodi, A.: RFI in SMOS Measurements: Update on Detection, Localization, Mitigation Techniques and Preliminary Quantified Impacts on Soil Moisture Products, 2014.

Rodriguez-Fernandez, N., Anterrieu, E., Boutin, J., Supply, A., Reverdin, G., Alory, G., Remy, E., Picard, G., Pellarin, T., Richaume, P., Mialon, A., Khazaal, A., Bitar, A. A., Rodriguez-Suquet, R., Yu, L., Gonzalez, P., Cheymol, C., Amiot, T., Maisongrande, P., Jeannin, N., Decoopman, T., Kallel, A., Morel, J.-M., Colom, M., Dunitz, M., Thouvenin-Masson, C., Olivier, L., and Kerr, Y. H.: The SMOS-HR Mission: Science Case and Project Status, in: IGARSS 2022 - 2022 IEEE International Geoscience and Remote Sensing Symposium, pp. 7182–7185, IEEE, Kuala Lumpur, Malaysia, ISBN 978-1-66542-792-0, doi:10.1109/IGARSS46834.2022.9883205, 2022.

Rodriguez-Fernandez, N., Rixen, T., and Boutin, J.: The Fine Resolution Explorer for Salinity, Carbon and Hydrology (FRESCH). A Mission to Study Ocean-Land-Ice Interfaces, in: IGARSS 2024 - 2024 IEEE International Geoscience and Remote Sensing Symposium, 2024.

Roy, A., Royer, A., Derksen, C., Brucker, L., Langlois, A., Mialon, A., and Kerr, Y. H.: Evaluation of Spaceborne L-Band Radiometer Measurements for Terrestrial Freeze/Thaw Retrievals in Canada, IEEE Journal of Selected Topics in Applied Earth Observations and Remote Sensing, 8, 4442–4459, doi:10.1109/JSTARS.2015.2476358, 2015.

Roy, A., Leduc-Leballeur, M., Picard, G., Royer, A., Toose, P., Derksen, C., Lemmetyinen, J., Berg, A., Rowlandson, T., and Schwank, M.: Modelling the L-Band Snow-Covered Surface Emission in a Winter Canadian Prairie Environment, Remote Sensing, 10, 1451, doi:10.3390/rs10091451, 2018.

Schwank, M., Stahli, M., Wydler, H., Leuenberger, J., Matzler, C., and Fluhler, H.: Microwave L-band Emission of Freezing Soil, IEEE Transactions on Geoscience and Remote Sensing, 42, 1252–1261, doi:10.1109/TGRS.2004.825592, 2004.

Schwank, M., Matzler, C., Wiesmann, A., Wegmuller, U., Pulliainen, J., Lemmetyinen, J., Rautiainen, K., Derksen, C., Toose, P., and Drusch, M.: Snow Density and Ground Permittivity Retrieved from L-Band Radiometry: A Synthetic Analysis, IEEE Journal of Selected Topics in Applied Earth Observations and Remote Sensing, 8, 3833–3845, doi:10.1109/JSTARS.2015.2422998, 2015.

Ulaby, F. and Long, D.: Microwave Radar and Radiometric Remote Sensing, University of Michigan Press, ISBN 978-0-472-11935-6, doi:10.3998/0472119356, 2014.

---

## Referee Report (RR1)

The new version shows improvements. But I still miss a clearer statement on deficiencies in the retrievals due to the temporal variation of areas with liquid water, and how this problem may be solved. Some information may even exist in the present data. By far the best results were obtained during the short period of available data in 2016 (Figure C1) for Inigok. Also in other years, the spring data seem to perform better than during the winter season.

Figures 5 to 7 include data that are not defined: the tiny points (very small rings).
The captions in these figures refer to other figures (in Figure 7 erroneously to Figure 7 again). I think that such information belongs in the main text. The figure caption should define the symbols, curves etc. of the associated figure, only, to avoid confusion, and to make it simpler for reading. In the caption to Figure 7, the reference to the x axis should  be: "probing depth below the soil surface".

It seems to me that the English language could be improved.

---

## Author Response (AR2)

*Retrieving frozen ground surface temperature under the snowpack in*
*Arctic permafrost area from SMOS observations*

Comments from Reviewer 2 / Christian Matzler

Comments from the reviewer
Answers from the author

─────────────────────────────────────────────────────

The authors want to thank again Christian Matzler for the valuable comments.

'The new version shows improvements. But I still miss a clearer statement on deficiencies in the retrievals due to the temporal variation of areas with liquid water, and how this problem may be solved. Some information may even exist in the present data. By far the best results were obtained during the short period of available data in 2016 (Figure C1) for Inigok. Also in other years, the spring data seem to perform better than during the winter season.

We consider that the "*areas with liquid water*" you mention refer to the open water / water bodies / lakes area. The subsection "5.2.1 Effects of the snow and ice-covered water bodies" is meant to highlight the spatial and temporal variability of water bodies and the faced limitations in our modeling. The use of the Hr,wi parameter is proposed as a "fixing" solution. The subsection "5.2.2 Analysis of a site with high water fraction (Inigok)" expands the discussion by looking at a specific site (Inigok). We added a comment about your relevant observation: "It is worth noting that the $\Delta \bar{T}$ are smaller in late winter, suggesting that early winter lakes freezing processes require a different dedicated modeling." (lines 404-406).

Figures 5 to 7 include data that are not defined: the tiny points (very small rings). The captions in these figures refer to other figures (in Figure 7 erroneously to Figure 7 again). I think that such information belongs in the main text. The figure caption should define the symbols, curves etc. of the associated figure, only, to avoid confusion, and to make it simpler for reading. In the caption to Figure 7, the reference to the x axis should be: "probing depth below the soil surface".'

Many thanks for pointing out the residual inconsistencies and typos.